# SAFETY SUBSPACES ARE NOT LINEARLY DISTINCT: A FINE-TUNING CASE STUDY

**Kaustubh Ponkshe*[1], Shaan Shah*[2], Raghav Singhal*[1], Praneeth Vepakomma[1,3]**
[1]Mohamed bin Zayed University of Artificial Intelligence
[2]University of California San Diego  [3]Massachusetts Institute of Technology

## ABSTRACT

Large Language Models (LLMs) rely on safety alignment to produce socially acceptable responses. However, this behavior is known to be brittle: further fine-tuning, even on benign or lightly contaminated data, can degrade safety and reintroduce harmful behaviors. A growing body of work suggests that alignment may correspond to identifiable directions in weight space, forming subspaces that could, in principle, be isolated or preserved to defend against misalignment. In this work, we conduct a comprehensive empirical study of this perspective. We examine whether safety-relevant behavior is concentrated in specific linear subspaces, whether it can be separated from general-purpose learning, and whether harmfulness arises from distinguishable patterns in activations. Across both weight and activation spaces, our findings are consistent: subspaces that amplify safe behaviors also amplify useful ones, and prompts with different safety implications activate overlapping representations. Rather than residing in distinct directions, we show that safety is highly entangled with the general learning components of the model. This suggests that subspace-based defenses face fundamental limitations and underscores the need for alternative strategies to preserve safety under continued training. We corroborate these findings with multiple experiments on five open-source LLMs from the Llama and Qwen families. Our code is publicly available at: https://github.com/CERT-Lab/safety-subspaces.

## 1 INTRODUCTION

Large Language Models (LLMs) show strong performance across a wide range of general-purpose tasks (1; 51; 60; 61; 63; 70). To ensure these models behave responsibly and align with human values, they undergo an additional process of *safety alignment*. This alignment is typically achieved during the post-training stage, enabling models to improve response quality, and refuse harmful prompts over the pre-trained stage. Despite known jailbreak methods that can bypass safeguards, aligned models are generally considered significantly safer than their base versions (44; 52; 67).

However, this alignment is fragile. Since safety is encoded in the model's weights, any modification, such as further fine-tuning (FT), can compromise it. This exposes a deeper attack surface beyond prompt engineering: an adversary could insert a small number of malicious samples into a training set to subvert alignment (4; 71; 72; 76). Recent work shows that even benign FT, low-rank adaptation, or pruning can degrade a model's safety profile (16; 17; 36; 50; 66). A growing line of research seeks to leverage the learned safety adherence to design defenses against adversarial attacks and interpret alignment mechanisms(3; 11; 20; 28; 33; 37; 69).

Fine-tuning (FT) is used to adapt an LLM to new and personalized domains and plays a key role in the widespread adoption of LLMs across diverse contexts. Preserving alignment in this setting, while retaining the improvements achieved through FT, is therefore both a practical concern and a technically challenging problem. This raises a natural question: *Does there exist a subspace, whether in weight space or activation space, that uniquely encodes safety alignment information without affecting performance?* If such a property exists, it could, in principle, enable the preservation of safety while maintaining model performance under continued training.

---

* denotes equal contribution. Author order decided randomly.

To construct defenses, prior works (20; 37) have typically derived safety subspaces using one of two approaches: weight updates from general alignment (aligned–base model deltas) or updates from targeted safety tuning (safety–base model deltas). Our goal is to comprehensively investigate these so-called "safety subspaces" in order to determine whether the information they contain is truly specific to safety. If so, we could separate unsafe information from the useful knowledge learned during FT through simple projections, thereby ensuring that our fine-tuned models are both robust to safety and high-performing.

To explore this question, we design four experiments probing the geometry of safety-related behavior across both model weights and activations. We begin by analyzing FT updates derived from purely useful and purely harmful datasets. These updates are projected into the candidate "safety subspaces" to test whether harmful updates are more expressive than useful ones within these subspaces. Next, we design an experiment involving contaminated FT, where a small fraction of harmful samples is mixed into an otherwise benign dataset. By projecting updates into the orthogonal complement of the candidate subspaces, we test whether harmful components can be selectively removed. From these experiments, we conclude that the candidate subspaces are not safety-specific but instead capture general learning. This leads us to ask whether any distinct safety subspace exists at all. To address this, our third experiment performs pairwise comparisons among useful, harmful, and safety updates to determine which pairs share the greatest similarity. Surprisingly, the similarity between harmful and safety updates is never the highest, as one might expect, and is sometimes even the lowest. Finally, in our fourth experiment, we extend this analysis to activation space, examining whether safety-specific attributes are distinguishable in activations rather than weights.

Across all experiments, we find no evidence that any linear subspace-whether in weight or activation space-captures safety-specific behavior in isolation. Although certain subspaces, such as those derived from the principle components of alignment or safety-specific updates, are impactful, they amplify both safe and useful behaviors alike, indicating that safety is deeply entangled with general learning. Similarly, activations from harmful and helpful prompts occupy overlapping regions of activation space, providing no evidence for distinct safety-related regions. Together, these findings reveal a fundamental limitation of linear subspace-based strategies. Since safe and harmful behaviors cannot be cleanly separated linearly, then projection- or filtering-based defenses are unlikely to suppress harmfulness without incurring comparable losses in utility. Our key contributions are as follows:

- We show that subspaces derived from alignment and safety-specific updates are not uniquely tied to safety; instead, they amplify both useful and harmful behaviors alike, implying that safety is deeply entangled with general learning (Section 3).
- We demonstrate that safety and harmful updates share no relatively significant subspace overlap, confirming that no region of weight space can be isolated specifically for safety (Section 5).
- Finally, we reaffirm this hypothesis in activation space, showing that harmful prompts do not activate distinct linear regions We observe that safety and harmful updates do not exhibit relatively higher within-task subspace overlap than cross-task comparisons, providing no evidence for a weight-space region that isolates safety from general learning signals (Section 6).
- Across multiple experiments on five open-source LLMs from the Llama and Qwen families, we consistently observe patterns inconsistent with linear separability of safety alignment, pointing to inherent limitations in subspace-based defenses.

## 2 PRELIMINARIES

**Notation.** Let $\mathbf{W}_0$ denote the parameters of the *base* model and $\mathbf{W}_A$ that of the *aligned* model. We denote the parameters of the model after *safety tuning*, i.e., fine-tuning specifically for harmless responses and refusals, using $\mathbf{W}_S$. We further fine-tune the aligned and the safety-tuned models on a task-specific dataset $\mathcal{D}j$, where $j \in \{\text{Useful}, \text{Harmful}, \text{Contaminated}\}$, resulting in parameters $\mathbf{W}_{\text{FT},j}$. We decompose the total parameter update as the sum of two components:

$$\Delta_A := \mathbf{W}_A - \mathbf{W}_0 \qquad \Delta_S := \mathbf{W}_S - \mathbf{W}_0 \qquad \text{(alignment/safety-specific updates)} \quad (1)$$

$$\Delta_T^j := \mathbf{W}_{\text{FT},j} - \mathbf{W}_A / \mathbf{W}_S \quad \text{(task-specific updates).} \qquad (2)$$

**Importance of Alignment Update ($\Delta_A$).** Alignment training typically emphasizes behavioral properties such as harmlessness, helpfulness, and honesty. Empirical studies (11; 20; 44) suggest

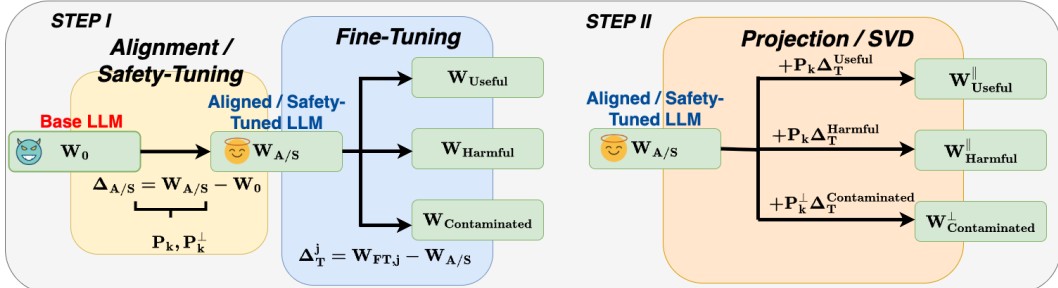

Figure 1: The base model $W_0$ is aligned/safety-tuned to produce the model $W_{A/S}$. **Step 1:** The difference $\Delta_{A/S} = W_{A/S} - W_o$ defines an alignment/safety-specific direction, from which projection matrices $P_k$ (top-K subspace) and $P_k^\perp$ (orthogonal subspace) are derived. $W_{A/S}$ is then fine-tuned on three datasets: helpful, harmful, and contaminated, to yield $W_{\text{useful}}$, $W_{\text{harmful}}$, and $W_{\text{contaminated}}$, with updates $\Delta_{t_j}$. **Step 2:** Project $\Delta_{t_j}$ using $P_k$ and $P_k^\perp$, and add back to $W_{A/S}$ to obtain projected models for evaluation. In addition, SVD is performed on the task-specific updates, and the Mode Subspace Overlap (MSO) is computed between the top-K singular vectors.

that the alignment update $\Delta_A$ encodes directions in parameter space that are strongly correlated with these safety attributes. This stage is also the sole point in production model training where safety is explicitly introduced into the model. Our goal is to systematically control the extent to which the subsequent task-specific update $\Delta_T^j$ interacts with these alignment directions.

**Importance of Safety-Specific Update ($\Delta_S$).**   We also aim to capture safety more directly, disentangling it from the broader behavioral changes introduced during the alignment stage. Safety tuning focuses explicitly on refusal and harmlessness, without simultaneously shaping general instruction-following ability. A subtle but important detail is that we use a distribution for safety tuning that is distinct from the one used for harmful fine-tuning, ensuring that our analysis captures genuine interactions rather than artifacts of dataset overlap. Our objective here is to systematically analyze how subsequent task-specific updates $\Delta_T^j$ interact with these safety directions.

**Constructing the Alignment & Safety Subspaces.**   To construct the alignment and safety subspaces, each tensor in the updates $\Delta_A/\Delta_S$ is first reshaped into a matrix (flattened if needed) $V_A, V_S \in \mathbb{R}^{M \times N}$. From here on, we use $V_{A/S}$ to refer to both $V_A, V_S$. We perform a thin singular value decomposition (SVD) of the form $V_{A/S} = U\Sigma V^\top$, which reveals the principal directions of parameter change (13; 43), ranked by their contribution to the Frobenius norm. The top $k$ (**Top-K**) right singular vectors in $V$ are then selected to define the *alignment/safety-specific subspace*:

$$\mathcal{S}_k := \text{span}(U_k), \quad U_k \in \mathbb{R}^{M \times k}, \quad k \leq \text{rank}(V_{A/S}). \tag{3}$$

Intuitively, $\mathcal{S}_k$ captures the $k$ most significant directions of parameter shifts resulting from alignment or safety-specific training. The subspaces naturally induce projection operators:

$$P_k := U_k U_k^\top, \quad P_k^\perp := I - P_k, \tag{4}$$

where $P_k$ projects a matrix onto the alignment/safety-specific subspace, and $P_k^\perp$ onto its orthogonal complement.

**Projection Schemes.**   Given a fractional rank hyperparameter $\varrho \in (0, 1]$, we determine $k = \lfloor \varrho \cdot \min(M, N) \rfloor$ and apply one of two projection-based update schemes to the task-specific update:

$$\textbf{Parallel}: \quad \tilde{\Delta}_T^j = P_k \Delta_T^j, \qquad\qquad \mathbf{W}_{\text{parallel}} = \mathbf{W}_{A/S} + \tilde{\Delta}_T^j, \tag{5}$$

$$\textbf{Orthogonal}: \quad \tilde{\Delta}_T^j = P_k^\perp \Delta_T^j, \qquad\qquad \mathbf{W}_{\text{orthogonal}} = \mathbf{W}_{A/S} + \tilde{\Delta}_T^j. \tag{6}$$

Eqn. 5 retains the update components that align with the candidate safety directions, while Eqn. 6 removes this component, retaining only the update orthogonal to the candidate safety subspace. Figure 1 provides an overview of our process.

**Control Experiments.** To further assess the specificity and effectiveness of the chosen safety subspace, we introduce two control experiments:

- **Random-K:** Instead of using the top-$k$ singular vectors from the SVD of $V_{A/S}$, we randomly sample $k$ singular vectors from the full set to construct a randomized safety subspace.
- **Random:** We replace $V_{A/S}$ with a random matrix of the same dimensions, perform its SVD, and use the top-$k$ singular vectors to define a synthetic safety subspace.

**Energy-Kept Ratio.** We introduce the fractional energy metric to quantify the extent of overlap between the task update and the safety subspace:

$$\mathcal{E}_k(\Delta_T^j) := \frac{\|P_k \Delta_T^j\|_F^2}{\|\Delta_T^j\|_F^2}, \quad \mathcal{E}_k^{\perp}(\Delta_T^j) = 1 - \mathcal{E}_k(\Delta_T^j). \tag{7}$$

**Mode Subspace Overlap (MSO).** Let $\mathbf{V} \in \mathbb{R}^{d \times n_V}$ and $\mathbf{W} \in \mathbb{R}^{d \times n_W}$ be two matrices with a shared ambient dimension $d$ but possibly different column counts. We extract their principal directions by taking the thin SVD:

$$\mathbf{V} = U_V \Sigma_V V_V^{\top}, \quad \mathbf{W} = U_W \Sigma_W V_W^{\top}. \tag{8}$$

For a chosen energy-retention fraction $\eta \in (0, 1]$, we select the smallest $k_V$ and $k_W$ such that the top $k_V$ (resp. $k_W$) left singular vectors capture at least an $\eta$-fraction of $\|\Sigma_V\|_F^2$ (resp. $\|\Sigma_W\|_F^2$). This yields orthonormal bases $Q_V \in \mathbb{R}^{d \times k_V}$ and $Q_W \in \mathbb{R}^{d \times k_W}$. The *overlap matrix* is then defined as:

$$S = Q_V^{\top} Q_W \in \mathbb{R}^{k_V \times k_W}. \tag{9}$$

To quantify the similarity between these $\eta$-energy subspaces, we use the MSO metric defined as:

$$\mathrm{MSO}(\mathbf{V}, \mathbf{W}; \eta) = \frac{\|S\|_F^2}{\min(k_V, k_W)}, \quad 0 \leq \mathrm{MSO} \leq 1. \tag{10}$$

Intuitively, $\mathrm{MSO}(\mathbf{V}, \mathbf{W}; \eta)$ measures the overlap between the top-$\eta$ energy components of $\mathbf{V}$ and $\mathbf{W}$: it equals 0 for orthogonal subspaces and 1 for identical spans. As a baseline, the expected overlap between random subspaces of dimensions $k_V$ and $k_W$ in $\mathbb{R}^d$ is given analytically by:

$$\mathbb{E}[\text{overlap}] = \frac{\max(k_V, k_W)}{d}. \tag{11}$$

**Models Used.** We evaluate base and aligned versions of five open-source LLMs: Llama 3.2 1B (12), Llama 2 7B (63), Qwen-2.5 1B (70), Qwen-2.5 3B, and Qwen-2.5 7B. To obtain safety-tuned variants, we fine-tune the base models on the safety-specific BeaverTails dataset (31), using only entries labeled `is_safe = True`.

## 3 DO ALIGNMENT SUBSPACES ENCODE SAFETY?

A central question in understanding safety alignment is whether specific directions in weight space, such as those defined by the difference between a base model and its RLHF-aligned counterpart, encode information unique to safety. If this is the case, constraining FT updates to lie within these subspaces could provide a principled approach to guarding against harmful optimization. We begin our investigation by examining whether task-specific FT updates align differently with the top directions of the alignment and safety-specific matrices, depending on whether the task is helpful or harmful.

**Experimental Setup.** We fine-tune the aligned and safety-tuned models on two distinct datasets. The first is a 20K subset of MetaMathQA (75), a benchmark of math word problems representing a useful task without safety concerns. The second is a 4K unsafe subset of BeaverTails (31), a synthetic dataset of harmful instruction–response pairs designed to elicit unsafe behavior. The resulting weight updates are denoted as $\Delta_T^{\text{Useful}}$ and $\Delta_T^{\text{Harmful}}$, respectively. To quantify behavioral effects, we evaluate harmfulness on AdvBench (80), with GPT-4o-mini (27) scoring each response from 1 (least harmful) to 5 (most harmful); the final score is the average across samples. Utility is measured as accuracy on the GSM8k test set (9), based on final-answer correctness. For each setting, we compute harmfulness, utility, and the energy-kept ratio for the projected models $\mathbf{W}_{\text{parallel}}$ and $\mathbf{W}_{\text{orthogonal}}$, as well as for the base, aligned, fine-tuned, and control models.

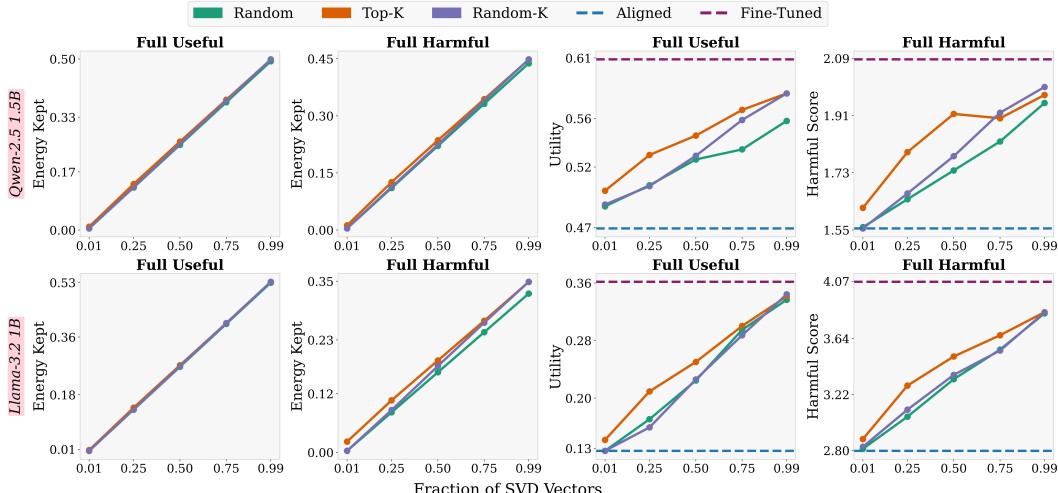

Figure 2: Parallel projection-based update schemes across varying SVD fractions. We report the energy-kept ratio for models fine-tuned on Full Useful and Full Harmful data, utility for models fine-tuned on Full Useful, and harmfulness for models fine-tuned on Full Harmful.

**Results: Energy Is Uniform Across Subspaces, Performance Is Not.**   As shown in Figures 2 and 6 (Appendix C), the fraction of energy retained in projected updates increases linearly with subspace rank and is consistent across all three subspace types. This pattern holds for both helpful and harmful updates. We find no evidence that update energy is preferentially concentrated in the top directions of $\Delta_{\mathrm{A/S}}$ for safe versus unsafe FT. This suggests that if a "safety subspace" exists, it is not revealed simply through energetic alignment with the dominant directions of $\Delta_{\mathrm{A/S}}$. At the same time, while energy is evenly distributed, behavioral impact is not. Figure 2 and Table 1 show that projecting $\Delta_{\mathrm{T}}^{\mathrm{Useful}}$ onto the top-$k$ directions consistently improves utility relative to random projections with equal energy. Similarly, projecting $\Delta_{\mathrm{T}}^{\mathrm{Harmful}}$ onto the same directions increases harmfulness. Thus, the top singular directions of $\Delta_{\mathrm{A/S}}$ are not uniquely aligned with safety, but they are generally potent: updates along these directions are more effective, whether the goal is to enhance utility or to elicit harmful behavior. Comprehensive results for all models are provided in Table 4 (Appendix C).

Table 1: Parallel projection-based update schemes across varying SVD fractions. We report the utility for models fine-tuned on Full Useful data, and harmfulness for models fine-tuned on Full Harmful.

| Model | Method | Utility (↑) | | | | | | | Harmful Score (↓) | | | | | | |
|---|---|---|---|---|---|---|---|---|---|---|---|---|---|---|---|
| | | Aligned | SVD Fractions | | | | | FT | Aligned | SVD Fractions | | | | | FT |
| | | | 0.01 | 0.25 | 0.50 | 0.75 | 0.99 | | | 0.01 | 0.25 | 0.50 | 0.75 | 0.99 | |
| | Top-K | 0.47 | 0.50 | 0.53 | 0.55 | 0.57 | 0.58 | 0.61 | 1.55 | 1.62 | 1.80 | 1.92 | 1.90 | 1.97 | 2.09 |
| Qwen-2.5 1.5B | Random-K | 0.47 | 0.49 | 0.50 | 0.53 | 0.56 | 0.58 | 0.61 | 1.55 | 1.55 | 1.66 | 1.78 | 1.92 | 2.00 | 2.09 |
| | Random | 0.47 | 0.49 | 0.50 | 0.53 | 0.53 | 0.56 | 0.61 | 1.55 | 1.56 | 1.65 | 1.74 | 1.83 | 1.95 | 2.09 |
| | Top-K | 0.13 | 0.14 | 0.21 | 0.25 | 0.30 | 0.34 | 0.36 | 2.80 | 2.89 | 3.29 | 3.51 | 3.66 | 3.84 | 4.07 |
| Llama-3.2 1B | Random-K | 0.13 | 0.13 | 0.16 | 0.23 | 0.29 | 0.34 | 0.36 | 2.80 | 2.83 | 3.11 | 3.37 | 3.55 | 3.84 | 4.07 |
| | Random | 0.13 | 0.13 | 0.17 | 0.22 | 0.29 | 0.34 | 0.36 | 2.80 | 2.81 | 3.05 | 3.34 | 3.56 | 3.83 | 4.07 |

**Implications: Alignment Directions Reflect General Learning, Not Safety.**   This symmetry across tasks is important. The fact that top-$k$ directions amplify both helpful and harmful behaviors equally suggests they do not encode alignment directly. Instead, they represent axes of general parameter sensitivity, ie. directions where updates tend to induce large changes in model behavior. This holds for both alignment and safety-specific updates, implying that disentangling safety offers no clear separation or benefit. In this sense, $\Delta_{\mathrm{A/S}}$ captures a general learning geometry: directions that are highly effective for optimization but not inherently safe. We draw three key takeaways. First, neither helpful nor harmful updates preferentially align with the top subspaces of $\Delta_{\mathrm{A}}$ or $\Delta_{\mathrm{S}}$ in terms of energy. Second, these same subspaces are more behaviorally expressive, amplifying both utility and harmfulness depending on the task. Third, this challenges the assumption that $\Delta_{\mathrm{A/S}}$ encode

safety-specific information expressed in their top subspaces. Thus, using $\Delta_{A/S}$ to constrain updates regulates the magnitude of behavioral change, but not its ethical nature.

# 4 CAN HARMFUL SUBSPACES BE REMOVED?

Having analyzed helpful and harmful updates in isolation, we now turn to a more realistic scenario: contaminated FT. This setting involves adding a small fraction of harmful examples to an otherwise benign dataset, producing updates that blend both signals. Prior work has shown that even limited contamination can erode safety, causing models to revert to unsafe behaviors (4; 36; 50; 72; 76). While earlier experiments identified expressive subspaces, we now ask the reverse question: can harmful components of an update be removed? We test whether filtering specific subspaces, particularly those aligned with the dominant directions of the alignment or safety-tuned matrix, can reduce harmfulness while preserving utility.

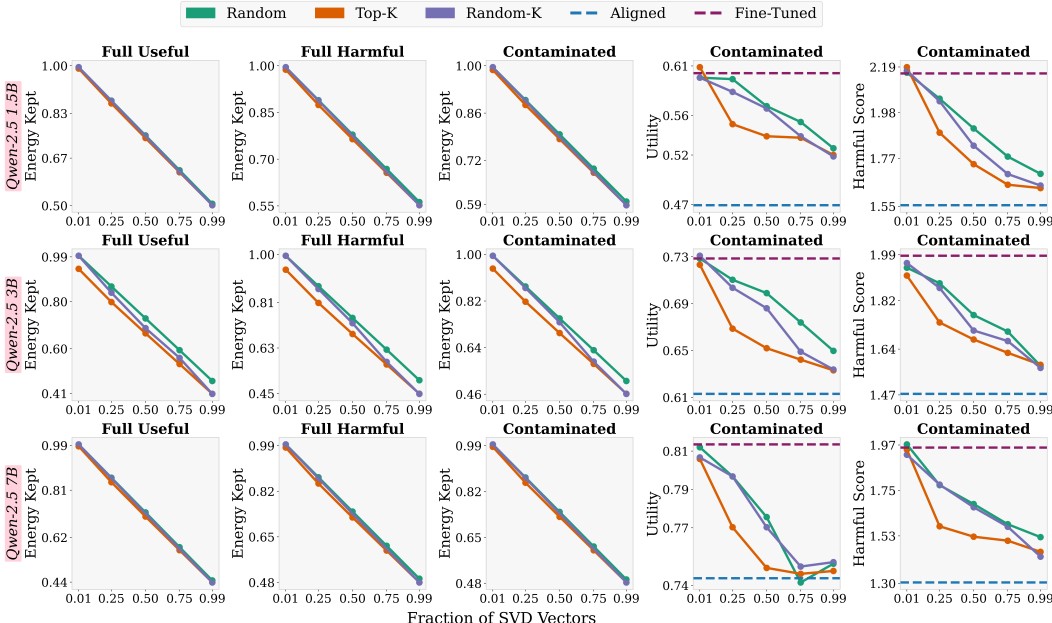

Figure 3: Orthogonal projection-based update schemes across varying SVD fractions. We report the energy-kept ratio for models fine-tuned on Full Useful, Full Harmful and Contaminated data; and utility and harmfulness for models fine-tuned on Contaminated.

**Experimental Setup.** We construct a contaminated dataset by mixing 20% harmful data from BeaverTails with 80% of the 20K MetaMathQA subset. FT on this mixture produces a single contaminated update, $\Delta_T$. To suppress harmful behavior, we apply the orthogonal projection strategy from Section 2, removing components along the top-$k$ alignment directions. Specifically, we compute $\tilde{\Delta}_T = P_k^{\perp}\Delta_T$, where $P_k^{\perp}$ projects onto the complement of the alignment subspace. We then evaluate the resulting models on GSM8K (utility) and AdvBench (harmfulness). Our objective is to test whether removing alignment-aligned components can reduce harmfulness while preserving task performance.

**Results: Utility And Harmfulness Drop Together.** Figures 3 and 7 (Appendix D) show the effects of orthogonal projection on retained energy, utility, and harmfulness. As $k$ increases, meaning more of the update is removed, retained energy declines steadily across all projection types (random, top-$k$, and random-$k$). Utility and harmfulness scores (Figure 3, Table 2) follow a similar downward trend. The rate of decline, however, differs by projection strategy. Removing top-$k$ alignment components reduces utility more sharply than random projections, while harmfulness decreases at a similar rate. This indicates no selective suppression of harmful behavior. In effect, safety gains come at a proportional cost to task performance, with no clear advantage in targeting the alignment subspace. Comprehensive results for all models are provided in Table 5 (Appendix D).

Table 2: Parallel projection-based update schemes across varying SVD fractions. We report the utility and harmfulness for models fine-tuned on Contaminated data.

| Model | Method | Utility (↑) | | | | | | | | Harmful Score (↓) | | | | | | | |
|---|---|---|---|---|---|---|---|---|---|---|---|---|---|---|---|---|---|
| | | Aligned | SVD Fractions | | | | | FT | | Aligned | SVD Fractions | | | | | FT | |
| | | | 0.01 | 0.25 | 0.50 | 0.75 | 0.99 | | | | 0.01 | 0.25 | 0.50 | 0.75 | 0.99 | | |
| Qwen-2.5 1.5B | Top-K | 0.47 | 0.50 | 0.53 | 0.55 | 0.57 | 0.58 | 0.60 | | 1.55 | 1.58 | 1.65 | 1.80 | 1.91 | 1.92 | 2.16 | |
| | Random-K | 0.47 | 0.49 | 0.52 | 0.53 | 0.55 | 0.55 | 0.60 | | 1.55 | 1.56 | 1.62 | 1.63 | 1.87 | 1.92 | 2.16 | |
| | Random | 0.47 | 0.49 | 0.50 | 0.52 | 0.52 | 0.54 | 0.61 | | 1.55 | 1.58 | 1.64 | 1.68 | 1.74 | 1.92 | 2.16 | |
| Qwen-2.5 3B | Top-K | 0.61 | 0.63 | 0.64 | 0.65 | 0.68 | 0.69 | 0.73 | | 1.47 | 1.49 | 1.58 | 1.69 | 1.76 | 1.83 | 1.99 | |
| | Random-K | 0.61 | 0.62 | 0.64 | 0.64 | 0.66 | 0.69 | 0.73 | | 1.47 | 1.45 | 1.55 | 1.62 | 1.65 | 1.91 | 1.99 | |
| | Random | 0.61 | 0.62 | 0.63 | 0.64 | 0.65 | 0.68 | 0.73 | | 1.47 | 1.45 | 1.50 | 1.57 | 1.75 | 1.83 | 1.99 | |
| Qwen-2.5 7B | Top-K | 0.74 | 0.74 | 0.75 | 0.75 | 0.75 | 0.78 | 0.81 | | 1.30 | 1.31 | 1.56 | 1.60 | 1.68 | 1.67 | 1.96 | |
| | Random-K | 0.74 | 0.74 | 0.75 | 0.76 | 0.75 | 0.78 | 0.81 | | 1.30 | 1.35 | 1.41 | 1.46 | 1.59 | 1.67 | 1.96 | |
| | Random | 0.74 | 0.74 | 0.75 | 0.75 | 0.75 | 0.78 | 0.81 | | 1.30 | 1.34 | 1.40 | 1.48 | 1.56 | 1.63 | 1.96 | |

**Implications: No Selective Removal Is Possible.** These results establish that the top subspaces of alignment or safety-tuned updates do not uniquely encode safety or harmfulness. Removing these directions degrades both utility and harmfulness at similar rates. If harmful behavior were confined to distinct subspaces, we would expect a steeper drop in harmfulness than in utility, yet this is not observed. Even if safety-relevant directions exist, they cannot be recovered from the alignment or safety-tuned matrices alone, particularly under contamination. The update blends helpful and harmful objectives, making its projection agnostic to intent. As a result, orthogonal projection fails to selectively suppress harmful behavior. Thus, subspace filtering based on alignment directions imposes a strict trade-off: improvements in safety come only at a proportional cost to utility.

We test whether projecting individual layers can cleanly separate safety and learning directions in Appendix I. Specifically, we repeat the two weight-space analyses described above and observe that no layer provides such separation.

## 5   ARE SAFETY WEIGHT SUBSPACES DISTINCT?

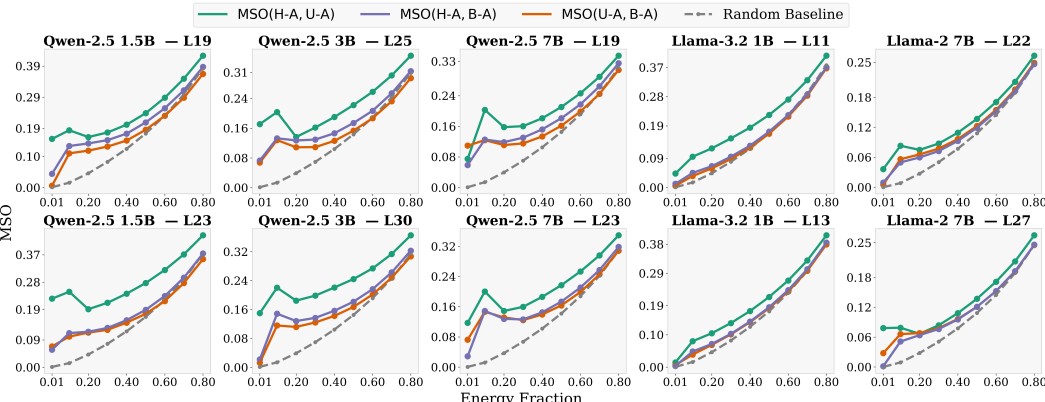

Figure 4: Mode Subspace Overlap (MSO) at the 70- and 85- percentile layers for pairwise comparisons of the dominant subspaces from Harmful fine-tuned (H), Aligned (A), and Base (B) models.

A natural question is whether a dedicated region of parameter space, which we might call a "safety subspace," captures safety-specific behavior. Such a subspace should satisfy two criteria: (i) safety-relevant updates, whether from alignment or harmful FT, should lie predominantly within it; and (ii) task-specific updates unrelated to safety should have minimal overlap, with projections onto the subspace leaving model safety unchanged. Our earlier results argue against the top subspaces of the alignment or safety-tuned matrices meeting these criteria. Nevertheless, it remains open whether *some other set of directions*, possibly outside these subspaces, could fulfill this role. To investigate this, we directly compare the dominant subspaces of different update types.

**Experimental Setup.** We compare the principal subspaces of three updates: the alignment update $\Delta_A$ (from the base to the aligned model), the harmful FT update $\Delta_T^{\text{Harmful}}$ (trained on BeaverTails), and the useful FT update $\Delta_T^{\text{Useful}}$ (trained on a 20K subset of MetaMathQA). Notably, the negated alignment update $-\Delta_A$ reverses alignment by pushing the model back toward its unaligned base state, effectively acting as a harmful update and serving as a useful reference point. We repeat these experiments for safety-tuned updates as well. For a given energy threshold $\eta \in (0, 1]$, we compute $\text{MSO}(\cdot, \cdot; \eta)$ (Section 2) for three pairs: (i) $\left(\Delta_T^{\text{Useful}}, \Delta_T^{\text{Harmful}}\right)$, to assess whether helpful and harmful FT affect similar subspaces; (ii) $\left(\Delta_T^{\text{Useful}}, -\Delta_A\right)$, to test the relationship between helpful updates and reversed alignment; and (iii) $\left(\Delta_T^{\text{Harmful}}, -\Delta_A\right)$, to compare two harmful directions. We sweep over $\eta$, with smaller values isolating high-energy directions and larger values approaching full-rank overlap. As a baseline, we include the random-subspace expectation $\max(k_V, k_W)/d$; values above this baseline indicate significant geometric alignment, while values near it suggest chance-level overlap.

**Results: Representations Overlap Across Tasks.** Figures 4 and 8 (Appendix E) show the pairwise overlap between the dominant subspaces (top-$k$ directions) of each update.

We report per-layer results in Figures 10, 11, 12, 13, and 14 (Appendix G), all of which show results consistent with our observations.

All pairs exhibit greater overlap than random baselines, indicating shared structure. However, in Figure 4, the strongest overlap is between the useful and harmful updates, rather than between alignment and harmful updates, as one might expect if safety were a shared component. This is a key finding. If a safety subspace existed, it would likely appear in the shared directions between alignment and harmful updates (or between safety and harmful updates), which affect safety in opposite ways. The absence of such overlap suggests that no consistent, linear safety-specific subspace exists. For safety-tuned models (Figure 8), the strongest overlap occurs between the useful and safety-specific updates, an even more counterintuitive result. This implies that, in terms of subspace overlap, the useful update lies closer to the safety-specific update than the harmful update does. Strikingly, this overlap is much larger than that between the harmful and safety-specific updates, even though, semantically, one might expect the latter to be most similar.

**Implications: Shared Weight Subspaces Drive Behavior, Not Safety.** Taken together, our results suggest that safety-relevant updates do not reside in a well-defined or isolatable subspace. Instead, alignment, safety, and harmfulness operate over complex, task-dependent directions. The strong overlap between harmful and helpful update subspaces indicates that these directions form a general *learning subspace*, expressive across tasks but agnostic to safety. Thus, we find no evidence of a distinct safety subspace, and linear subspace methods cannot cleanly isolate safety in parameter space. This highlights a fundamental limitation in subspace-based defences: attempts to filter safety-relevant components suppress general learning as well.

We repeat all of our weight-space analyses for a model safety-tuned using only Direct Preference Optimization (DPO) in Appendix H, to assess whether our findings remain consistent across different safety training strategies. We find that all our results transfer cleanly to this setting.

## 6 DO SAFETY SUBSPACES EXIST IN ACTIVATION SPACE?

So far, our analysis has focused on weight space, probing whether certain update directions encode safety-related behavior. Finding no evidence of distinct safety subspaces at the parameter level raises a final question: do safety-relevant inputs elicit distinct activation patterns, even when their weight updates overlap? Although weight updates may distribute broadly, inputs might still selectively activate specific directions. We investigate this possibility in the following section.

**Experimental Setup.** We compare internal activations induced by different prompt categories. Specifically, we pass useful (benign) prompts from the MATH dataset (19) and harmful prompts from BeaverTails (test set) and ToxiGen (15) through three models: the aligned model, the useful fine-tuned model, and the contaminated fine-tuned model. For each prompt, we record the hidden state of the *last* generated token (35) at each transformer layer $\ell \in 0, \ldots, L$. At each layer, these hidden states are stacked into activation matrices of shape $\mathbb{R}^{n \times d}$, where $d$ is the hidden size and $n$ is

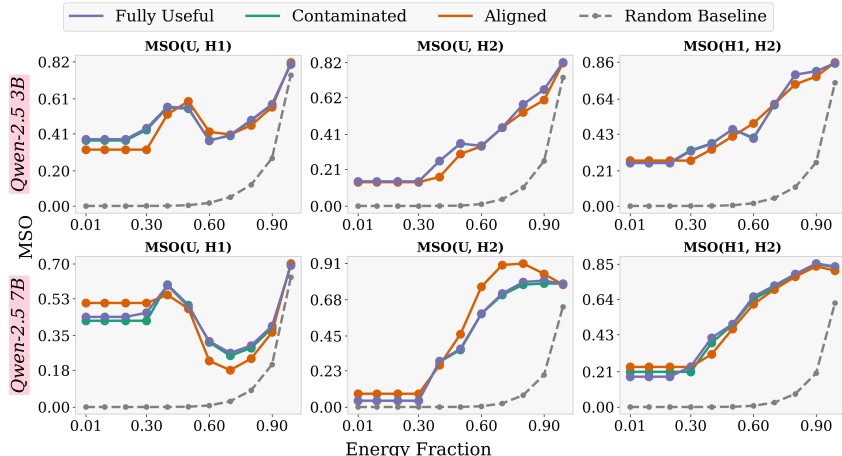

Figure 5: Average Mode Subspace Overlap (MSO) across layers in the 65–90% depth range for pairwise comparisons of activations from Useful (U) and multiple Harmful (H1, H2) prompt sets.

the number of prompts (5000 per dataset). We then compute MSO (see Section 2) between activation matrices from different datasets, sweeping over energy thresholds $\eta$. Smaller values of $\eta$ capture high-energy activation modes, while larger values approximate full-rank comparisons. We plot MSO curves alongside the random-subspace baseline $\max(k_{\text{Useful}}, k_{\text{Harmful}})/d$ and report averages over layers in the 65–90% depth percentile.

**Results: Representation Subspaces Overlap Across Tasks.** Figure 5 shows MSO values across all pairs of prompt categories. Useful and harmful prompts consistently exhibit overlap above the random baseline, indicating activation of shared high-energy subspaces in activation space. Interestingly, the overlap between the two harmful prompt sets is not always greater than their overlap with useful prompts; in some cases, the useful–harmful overlap exceeds the harmful–harmful overlap. The degree of overlap also varies across model configurations. Some models show strong alignment even in the top subspaces, while others exhibit more gradual increases, with overlap becoming significant only at higher energy thresholds. This variability suggests that representational similarity depends more on model-specific factors than on the safety content of the prompts. Additional results on other models are provided in Figure 9 (Appendix F). We report additional analysis experiments in Appendix K.

**Implications: Shared Activation Subspaces Drive Behavior, Not Safety.** These observations suggest that while all prompt types activate shared subspaces more than expected by chance, there is no evidence of a distinct safety-violating subspace. If such a subspace existed, activations from harmful prompts would consistently show greater mutual overlap than with useful prompts, which is not observed. Instead, prompts with different safety implications are processed through broadly overlapping representations. This supports our earlier hypothesis: the directions most responsible for behavior correspond to general-purpose representational subspaces rather than safety-specific ones. These directions are activated across tasks and prompt types, indicating that LLMs do not internally separate "safe" and "unsafe" activation modes but instead rely on shared, high-impact subspaces. We therefore find no evidence of a distinct safety subspace even in activation space. Together with our weight-space results, this suggests that both aligned and harmful behaviors arise from shared representational mechanisms rather than separable subspaces.

## 7 CONCLUSION

This work set out to investigate how safety alignment is encoded in LLMs and whether it can be isolated in weight or activation space. Our findings challenge the common assumption that alignment or safety-specific updates correspond to unique "safety subspaces". Subspaces with strong behavioral impact are not unique to safety; rather, they amplify both utility and harmfulness, indicating that safety is deeply entangled with general learning. Similarly, harmful and useful prompts activate overlapping regions of activation space, providing no evidence for a distinct safety subspace. Together,

these results establish that safety alignment is not linearly separable in LLMs. While this complicates the development of subspace-based defenses, it also highlights the potential of high-impact directions, if appropriately constrained, for guiding both safe fine-tuning and activation-level control. More broadly, our work calls for rethinking assumptions in interpretability and alignment research, and for developing methods that explicitly account for the entangled nature of representations.

## REPRODUCIBILITY STATEMENT

We place a strong emphasis on reproducibility. To this end, we make our implementation publicly available at https://github.com/CERT-Lab/safety-subspaces. A complete description of the experimental setup is provided throughout the paper in Sections 3, 4, 5 and 6, with additional hyperparameter details in Appendix B. All experiments are conducted on widely used, publicly available benchmark datasets, which we summarize in Appendix M.

## ACKNOWLEDGEMENTS AND DISCLOSURE OF FUNDING

This work received support from Mohamed bin Zayed University of Artificial Intelligence (MBZUAI) and was partially funded by the ADIA Lab Fellowship.

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

# Appendix

CONTENTS

## A    RELATED WORK

**Safety Alignment and Task-Specific Fine-Tuning in LLMs.**    Large Language Models (LLMs) do not inherently follow instructions and often exhibit socially undesirable behaviors. To address this, various post-training methods, instruction-tuning and reinforcement learning from human feedback, are applied to align base LLMs with human values and improve their instruction-following capabilities (44; 52; 56; 67). However, studies have shown that fine-tuning these aligned models on harmful data can undo this alignment, restoring their original, socially unacceptable behaviors (71). This

unalignment phenomenon has been demonstrated in both open-source models (36; 72) and proprietary models (4; 50; 76) via publicly available fine-tuning APIs, thereby exposing a new attack surface (10; 25; 32). Moreover, even fine-tuning on benign downstream tasks can degrade alignment (16; 17).

**Defense Methods.** To safeguard aligned LLMs against unalignment during fine-tuning, defenses have been proposed at three stages of the pipeline: the alignment stage, the fine-tuning stage, and the post-processing stage. The effectiveness of these defense methods is evaluated using downstream model utility and harmfulness (23).

**Alignment Stage Defenses.** Alignment stage defenses update the initial instruction-tuning process to ensure that downstream fine-tuning cannot easily overwrite the model's safety behavior. One approach augments the alignment loss, making harmful representations harder to recover during fine-tuning updates (54). Another line of work relies on safety-oriented data curation to preserve alignment under downstream fine-tuning(40). Adversarial and meta-learning techniques have also been combined to develop tamper-resistant methods that prevent harmfulness while maintaining task performance (59). A separate strategy introduces a regularization term to the alignment loss, which has been shown to preserve safety after fine-tuning (22). Perturbing safety-critical layers during instruction-tuning has also been shown to protect alignment (39). Additional work traces unalignment to excessive dependence on maximum-likelihood training, motivating an integrity preserving variant of this method (7). A study on "shallow alignment" also shows that instruction-tuning influences only the first few output tokens, whereas deeper alignment improves robustness (49).

**Fine-Tuning Stage Defenses.** Fine-tuning stage defenses modify the fine-tuning process to ensure that the model's alignment is preserved after update. One class of defenses focuses on data curation, augmenting the fine-tuning dataset to maintain alignment after update (5; 14). Another approach uses safety examples prefixed with a secret prompt, which act as backdoor triggers to reactivate safe behavior after fine-tuning (64). A data ranking based strategy has also been proposed, where low-quality data is down-ranked and high-quality data is up-ranked to better preserve safety (58). It has also been shown that prompt templates play an important role; removing the safety prompt during fine-tuning and reintroducing it at inference time can maintain alignment (42).

Optimization based defenses are another type of fine-tuning stage defenses. One line of work splits fine-tuning into an alignment phase and a utility phase, safeguarding both safety and task performance (24). Another approach combines safety and helpfulness objectives into a single loss (77).

Parameter level methods can also be used to preserve safety. One strategy identifies safety neurons and updates only those parameters (78). Another approach involves localizing safety layers and freezing their gradients, which has been shown to prevent unalignment (38). Another line of work explores constraining parameter changes to directions orthogonal to existing safety features, showing that this method preserves alignment (37). It has also been shown that harmful data can be filtered by matching fine-tuning embeddings against the top-k singular vectors of an activation matrix generated using a harmful dataset (8).

**Post-Processing Stage Defenses.** Post-processing stage defenses adjust the fine-tuned model to restore alignment and preserve usefulness. One approach adds a safety vector, defined as the difference between aligned and unaligned weights, to the fine-tuned parameters to regain safe behavior (3). Another line of work projects the fine-tuning update onto the alignment vector when their similarity drops below a threshold, or selectively merges layers from the fine-tuned and aligned models under the same criterion to achieve a similar effect (11; 20). A third strategy removes parameters identified as harmful after fine-tuning to restore alignment (21). It has been shown that safety directions in attention-head activations can also be located and used for targeted intervention (79) to realign the fine-tuned model. Another method detects update parameters whose signs contradict the original alignment and removes them (69). Additional work restores/finds safety-critical neurons (6; 73), fuses aligned and fine-tuned models (74), or adds an optimized post-hoc perturbation to recover alignment (65).

**Safety Mechanisms in Fine-Tuned and Aligned LLMs.** Recent studies have examined how LLMs express safety over neurons, layers, and activations. One study finds that safety related information is language agnostic, identifies parameters whose modification affects alignment, and shows that

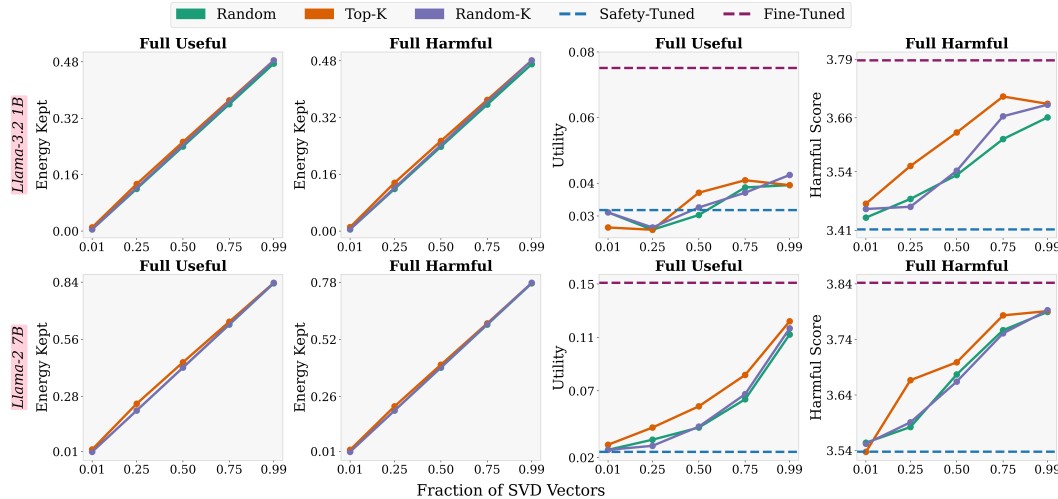

Figure 6: Parallel projection-based update schemes across varying SVD fractions. We report the energy-kept ratio for models fine-tuned on Full Useful and Full Harmful data, utility for models fine-tuned on Full Useful, and harmfulness for models fine-tuned on Full Harmful.

freezing these parameters during fine-tuning does not ensure safety (48). Another line of work locates sparse regions in parameter space whose removal weakens alignment, and likewise observes that freezing these regions alone is insufficient to maintain model alignment (66). A separate analysis maps a safety basin in weight space, noting that random perturbations inside the basin leave safety intact, whereas fine-tuning moves weights outside it (47). Work on the activation residual stream isolates a refusal direction, removing this direction prevents refusal to harmful prompts, while adding it triggers refusal to benign ones (2). Other work shows that safety in models is governed by multiple directions in residual stream space, and that refusal rates drop when prompts avoid tokens activating these directions (45). Finally, a study shows that safety fine-tuning minimally adjusts MLP weights by pushing unsafe inputs into the weights' null space, leading models to process adversarial prompts as safe (29).

## B  EXPERIMENTAL DETAILS

We implemented all experiments using PyTorch (46) and the HuggingFace Transformers library (68). We ran all experiments on a single NVIDIA A6000 GPU (48 GB). To save memory, all base models are initalized in **torch.bfloat16** precision. All models are trained using the AdamW optimizer (41). Detailed hyperparameter configurations for full fine-tuning (and safety-tuning) of each model are presented in Table 3.

Table 3: Hyperparameter settings for fine-tuning the various models.

| Optimizer | AdamW |
|---|---|
| Batch size | 1 |
| Max. Seq. Len | 512 |
| Grad Acc. Steps | 32 |
| Epochs | 1 |
| Learning Rate | $1 \times 10^{-5}$ |
| LR Scheduler | Cosine |
| Warmup Ratio | 0.02 |

## C  DO ALIGNMENT SUBSPACES ENCODE SAFETY?

We provide additional results in Figure 6 and Table 4 to support the analysis presented in Section 3.

Table 4: Parallel projection-based update schemes across varying SVD fractions. We report the utility for models fine-tuned on Full Useful data, and harmfulness for models fine-tuned on Full Harmful.

| Model | Method | Utility (↑) SVD Fractions | | | | | Harmful Score (↓) SVD Fractions | | | | |
|---|---|---|---|---|---|---|---|---|---|---|---|
| | | 0.01 | 0.25 | 0.50 | 0.75 | 0.99 | 0.01 | 0.25 | 0.50 | 0.75 | 0.99 |
| Qwen-2.5 1.5B | Base | | | 0.21 | | | | | 3.27 | | |
| | Aligned | | | 0.47 | | | | | 1.55 | | |
| | Fine-Tuned | | | 0.61 | | | | | 2.09 | | |
| | Top-K | 0.50 | 0.53 | 0.55 | 0.57 | 0.58 | 1.62 | 1.80 | 1.92 | 1.90 | 1.97 |
| | Random-K | 0.49 | 0.50 | 0.53 | 0.56 | 0.58 | 1.55 | 1.66 | 1.78 | 1.92 | 2.00 |
| | Random | 0.49 | 0.50 | 0.53 | 0.53 | 0.56 | 1.56 | 1.65 | 1.74 | 1.83 | 1.95 |
| Llama-3.2 1B | Base | | | 0.03 | | | | | 4.13 | | |
| | Aligned | | | 0.13 | | | | | 2.80 | | |
| | Fine-Tuned | | | 0.36 | | | | | 4.07 | | |
| | Top-K | 0.14 | 0.21 | 0.25 | 0.30 | 0.34 | 2.89 | 3.29 | 3.51 | 3.66 | 3.84 |
| | Random-K | 0.13 | 0.16 | 0.23 | 0.29 | 0.34 | 2.83 | 3.11 | 3.37 | 3.55 | 3.84 |
| | Random | 0.13 | 0.17 | 0.22 | 0.29 | 0.34 | 2.81 | 3.05 | 3.34 | 3.56 | 3.83 |
| Llama-3.2 1B | Base | | | 0.026 | | | | | 4.13 | | |
| | Safety-Tuned | | | 0.032 | | | | | 3.41 | | |
| | Fine-Tuned | | | 0.075 | | | | | 3.79 | | |
| | Top-K | 0.026 | 0.026 | 0.037 | 0.041 | 0.039 | 3.47 | 3.55 | 3.63 | 3.71 | 3.69 |
| | Random-K | 0.031 | 0.026 | 0.033 | 0.037 | 0.042 | 3.46 | 3.46 | 3.54 | 3.66 | 3.69 |
| | Random | 0.031 | 0.026 | 0.030 | 0.038 | 0.039 | 3.44 | 3.48 | 3.54 | 3.61 | 3.66 |
| Qwen-2.5 3B | Base | | | 0.44 | | | | | 2.53 | | |
| | Aligned | | | 0.61 | | | | | 1.47 | | |
| | Fine-Tuned | | | 0.72 | | | | | 2.16 | | |
| | Top-K | 0.63 | 0.64 | 0.65 | 0.68 | 0.69 | 1.48 | 1.71 | 1.81 | 1.91 | 1.92 |
| | Random-K | 0.62 | 0.63 | 0.64 | 0.65 | 0.69 | 1.44 | 1.55 | 1.62 | 1.74 | 1.91 |
| | Random | 0.62 | 0.63 | 0.64 | 0.65 | 0.68 | 1.44 | 1.50 | 1.66 | 1.75 | 1.83 |
| Qwen-2.5 7B | Base | | | 0.69 | | | | | 1.90 | | |
| | Aligned | | | 0.74 | | | | | 1.30 | | |
| | Fine-Tuned | | | 0.81 | | | | | 2.12 | | |
| | Top-K | 0.72 | 0.74 | 0.76 | 0.77 | 0.77 | 1.34 | 1.56 | 1.66 | 1.76 | 1.84 |
| | Random-K | 0.73 | 0.75 | 0.74 | 0.75 | 0.77 | 1.34 | 1.44 | 1.53 | 1.64 | 1.84 |
| | Random | 0.74 | 0.75 | 0.75 | 0.76 | 0.76 | 1.33 | 1.40 | 1.48 | 1.56 | 1.75 |
| Llama-2 7B | Base | | | 0.05 | | | | | 4.27 | | |
| | Aligned | | | 0.20 | | | | | 1.74 | | |
| | Fine-Tuned | | | 0.30 | | | | | 3.41 | | |
| | Top-K | 0.21 | 0.24 | 0.26 | 0.28 | 0.29 | 1.81 | 2.34 | 2.61 | 2.90 | 3.15 |
| | Random-K | 0.20 | 0.23 | 0.25 | 0.28 | 0.29 | 1.74 | 1.91 | 2.09 | 2.63 | 3.13 |
| | Random | 0.20 | 0.23 | 0.25 | 0.28 | 0.28 | 1.77 | 1.91 | 2.15 | 2.57 | 3.03 |
| Llama-2 7B | Base | | | 0.053 | | | | | 4.27 | | |
| | Safety-Tuned | | | 0.024 | | | | | 3.54 | | |
| | Fine-Tuned | | | 0.151 | | | | | 3.84 | | |
| | Top-K | 0.030 | 0.042 | 0.058 | 0.082 | 0.122 | 3.54 | 3.67 | 3.70 | 3.78 | 3.79 |
| | Random-K | 0.026 | 0.029 | 0.043 | 0.067 | 0.117 | 3.55 | 3.59 | 3.66 | 3.75 | 3.79 |
| | Random | 0.026 | 0.033 | 0.042 | 0.064 | 0.112 | 3.55 | 3.58 | 3.68 | 3.76 | 3.79 |

# D    CAN HARMFUL SUBSPACES BE REMOVED?

Figure 7 and Table 5 presents supplementary results that further substantiate the findings discussed in Section 4.

# E    ARE SAFETY WEIGHT SUBSPACES DISTINCT?

To supplement the analysis in Section 5, we report extended results in Figure 8.

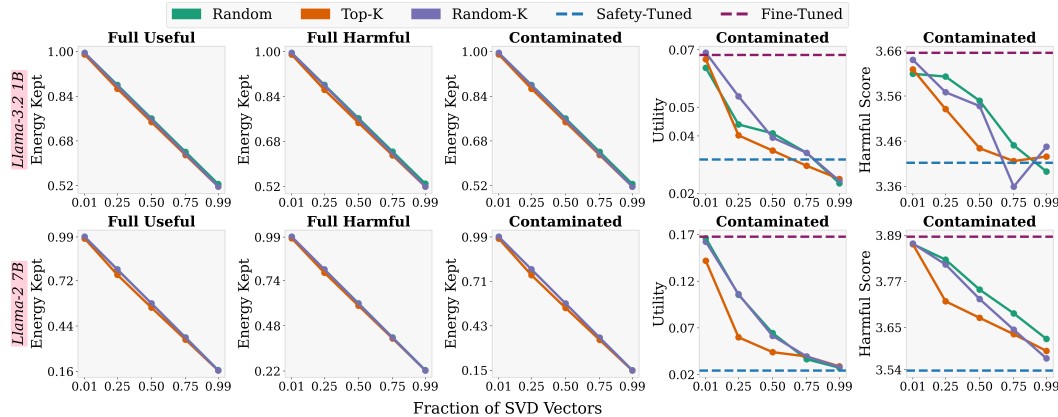

Figure 7: Orthogonal projection-based update schemes across varying SVD fractions. We report the energy-kept ratio for models fine-tuned on Full Useful, Full Harmful and Contaminated data; and utility and harmfulness for models fine-tuned on Contaminated.

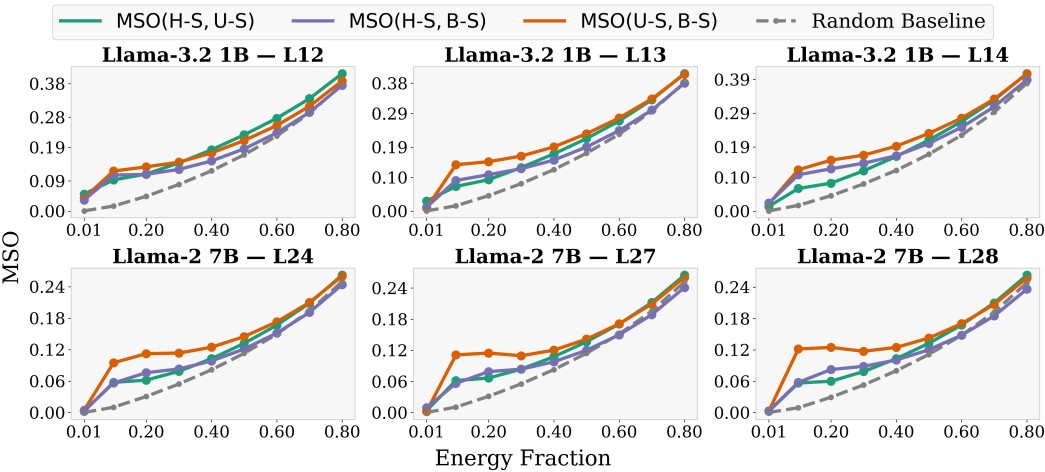

Figure 8: MSO at the 70-, 80-, and 85- percentile layers for pairwise comparisons of the dominant weight subspaces from Harmful fine-tuned (H), Safety-Tuned (S), and Base (B) models.

## F  DO SAFETY SUBSPACES EXIST IN ACTIVATION SPACE?

To complement the discussion in Section 6, we include extended results in Figure 9.

## G  CAN SAFETY WEIGHT SUBSPACES BE DISTINCT LAYERWISE?

To complete the analysis in Section 5, we report per-layer results in Figures 10, 11, 12, 13, and 14. We observe that the strongest overlap is **never** between the alignment and harmful updates, contrary to what one might expect if safety were represented as a shared component. This pattern is consistent across all layers and all models.

## H  EFFECT OF SAFETY-TUNING VIA DIRECT PREFERENCE OPTIMIZATION

In this section, we extend our study to include a different alignment method, specifically the widely-used Direct Preference Optimization (DPO, (53)). This allows us to study how overlap varies under a different alignment method. In doing so, we can determine whether the entanglement we observe is a fundamental property of the model's architecture or a consequence of the alignment procedure, and whether it persists across different forms of safety training.

Table 5: Parallel projection-based update schemes across varying SVD fractions. We report the utility and harmfulness for models fine-tuned on Contaminated data.

| Model | Method | Utility (↑) SVD Fractions | | | | | Harmful Score (↓) SVD Fractions | | | | |
|---|---|---|---|---|---|---|---|---|---|---|---|
| | | 0.01 | 0.25 | 0.50 | 0.75 | 0.99 | 0.01 | 0.25 | 0.50 | 0.75 | 0.99 |
| Qwen-2.5 1.5B | Base | | | 0.21 | | | | | 3.27 | | |
| | Aligned | | | 0.47 | | | | | 1.55 | | |
| | Fine-Tuned | | | 0.60 | | | | | 2.16 | | |
| | Top-K | 0.50 | 0.53 | 0.52 | 0.55 | 0.56 | 1.59 | 1.65 | 1.79 | 1.91 | 1.92 |
| | Random-K | 0.49 | 0.52 | 0.53 | 0.55 | 0.55 | 1.56 | 1.62 | 1.63 | 1.87 | 1.92 |
| | Random | 0.49 | 0.50 | 0.52 | 0.52 | 0.54 | 1.58 | 1.64 | 1.68 | 1.74 | 1.92 |
| Llama-3.2 1B | Base | | | 0.03 | | | | | 4.13 | | |
| | Aligned | | | 0.13 | | | | | 2.80 | | |
| | Fine-Tuned | | | 0.37 | | | | | 3.60 | | |
| | Top-K | 0.14 | 0.20 | 0.25 | 0.29 | 0.33 | 2.84 | 2.90 | 3.05 | 3.36 | 3.45 |
| | Random-K | 0.13 | 0.16 | 0.22 | 0.29 | 0.33 | 2.81 | 2.90 | 3.03 | 3.19 | 3.45 |
| | Random | 0.13 | 0.16 | 0.22 | 0.28 | 0.33 | 2.84 | 2.90 | 3.19 | 3.19 | 3.45 |
| Llama-3.2 1B | Base | | | 0.026 | | | | | 4.13 | | |
| | Safety-Tuned | | | 0.032 | | | | | 3.41 | | |
| | Fine-Tuned | | | 0.068 | | | | | 3.65 | | |
| | Top-K | 0.027 | 0.026 | 0.033 | 0.048 | 0.039 | 3.42 | 3.48 | 3.59 | 3.56 | 3.62 |
| | Random-K | 0.030 | 0.026 | 0.033 | 0.042 | 0.040 | 3.43 | 3.44 | 3.40 | 3.46 | 3.60 |
| | Random | 0.032 | 0.026 | 0.032 | 0.039 | 0.039 | 3.42 | 3.49 | 3.44 | 3.53 | 3.59 |
| Qwen-2.5 3B | Base | | | 0.44 | | | | | 2.53 | | |
| | Aligned | | | 0.61 | | | | | 1.47 | | |
| | Fine-Tuned | | | 0.73 | | | | | 1.99 | | |
| | Top-K | 0.62 | 0.63 | 0.65 | 0.68 | 0.69 | 1.49 | 1.58 | 1.69 | 1.76 | 1.83 |
| | Random-K | 0.62 | 0.64 | 0.64 | 0.66 | 0.69 | 1.45 | 1.55 | 1.62 | 1.65 | 1.91 |
| | Random | 0.62 | 0.63 | 0.64 | 0.65 | 0.68 | 1.45 | 1.50 | 1.57 | 1.75 | 1.83 |
| Qwen-2.5 7B | Base | | | 0.69 | | | | | 1.90 | | |
| | Aligned | | | 0.74 | | | | | 1.30 | | |
| | Fine-Tuned | | | 0.81 | | | | | 1.96 | | |
| | Top-K | 0.74 | 0.75 | 0.75 | 0.75 | 0.78 | 1.30 | 1.55 | 1.60 | 1.68 | 1.67 |
| | Random-K | 0.74 | 0.75 | 0.76 | 0.75 | 0.78 | 1.35 | 1.41 | 1.46 | 1.59 | 1.67 |
| | Random | 0.74 | 0.75 | 0.75 | 0.75 | 0.78 | 1.34 | 1.40 | 1.48 | 1.56 | 1.63 |
| Llama-2 7B | Base | | | 0.05 | | | | | 4.27 | | |
| | Aligned | | | 0.20 | | | | | 1.74 | | |
| | Fine-Tuned | | | 0.30 | | | | | 3.08 | | |
| | Top-K | 0.21 | 0.23 | 0.25 | 0.27 | 0.28 | 1.77 | 1.91 | 2.15 | 2.38 | 2.74 |
| | Random-K | 0.20 | 0.23 | 0.26 | 0.28 | 0.28 | 1.74 | 1.91 | 2.09 | 2.38 | 2.79 |
| | Random | 0.20 | 0.23 | 0.25 | 0.27 | 0.28 | 1.77 | 1.91 | 2.15 | 2.38 | 2.74 |
| Llama-2 7B | Base | | | 0.053 | | | | | 4.27 | | |
| | Safety-Tuned | | | 0.024 | | | | | 3.54 | | |
| | Fine-Tuned | | | 0.168 | | | | | 3.89 | | |
| | Top-K | 0.029 | 0.036 | 0.055 | 0.082 | 0.127 | 3.49 | 3.69 | 3.76 | 3.83 | 3.85 |
| | Random-K | 0.026 | 0.032 | 0.042 | 0.062 | 0.128 | 3.59 | 3.58 | 3.75 | 3.79 | 3.89 |
| | Random | 0.026 | 0.036 | 0.041 | 0.065 | 0.122 | 3.57 | 3.64 | 3.74 | 3.78 | 3.89 |

We repeat the full weight-space analyses from Sections 3, 4, and 5 for a Llama-3.2 1B model that we safety-tuned using DPO on the PKU-SafeRLHF dataset (30). We train the model for 5k steps on a 10k-example subset of the dataset.

The results are shown in Figures 15, 16, and 17. We observe that all of our previous hypotheses and analyses hold consistently across these experiments. Together, these findings suggest that the underlying geometric behavior we identify is robust to the choice of alignment method.

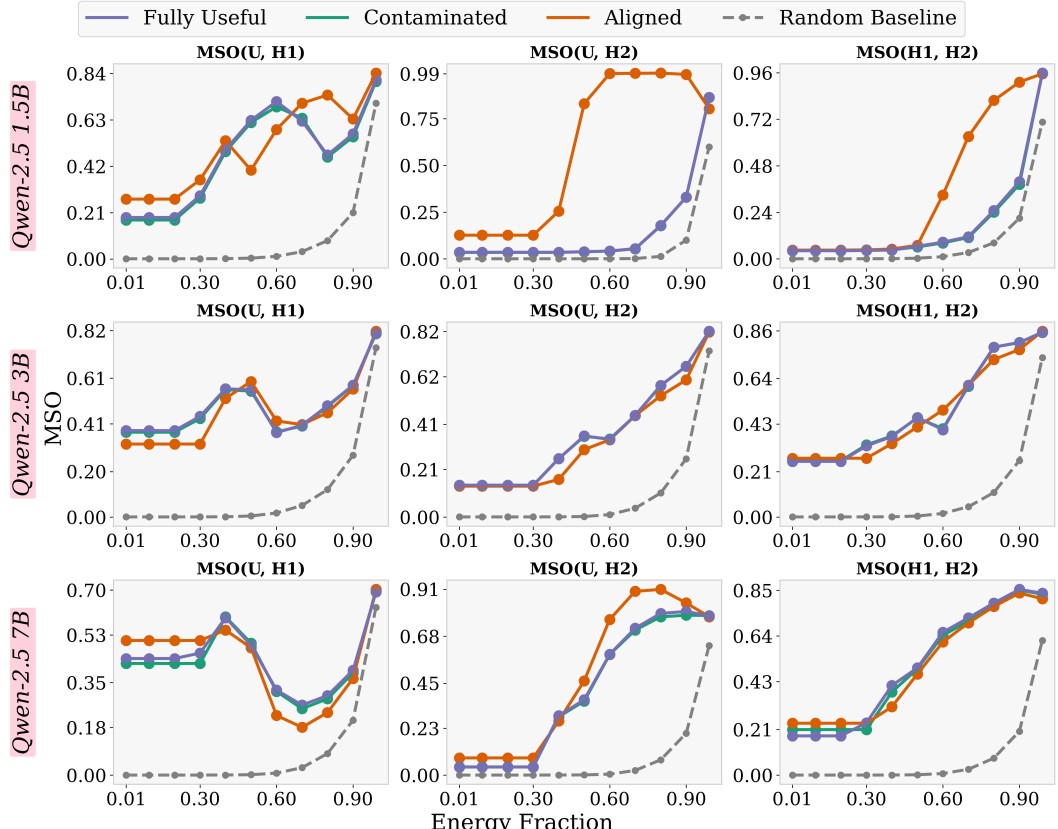

Figure 9: MSO across layers in the 65–90% depth range for pairwise comparisons of activations from Useful (U) and multiple Harmful (H1, H2) prompt sets.

# I   DO SAFETY AND LEARNING DECOUPLE AT ANY LAYER?

In this section, we investigate whether a layer-specific strategy exists for isolating safety and learning subspaces. Specifically, we ask whether projecting (k) components from a *single* layer yields different behavior than projecting the same number of components across *all* layers.

We test this on Qwen2.5-1.5B by repeating the experiments from Sections 3 and 4, but restricting the projection to individual layers. We evaluate approximately every third layer in the network, resulting in 10 distinct layers.

The results, shown in Figures 18 and 19, reinforce our earlier hypothesis: no individual layer exhibits a clean separation between safety and learning directions. This further strengthens our claim that safety and task-learning signals are geometrically entangled throughout the model.

# J   ROBUSTNESS CHECKS: GENERAL ABILITY AND OVER-REFUSAL

## J.1   ASSESSING GENERAL ABILITY CONFOUNDS VIA MMLU

We want to test whether the harmfulness metric is confounded with general ability (instruction following and coherence). More concretely, we aim to determine whether the results reported in our paper could simply be due to a loss in general capability. To evaluate this, we use MMLU (Massive Multitask Language Understanding, (18)), a widely used benchmark for general ability. We repeat the experiments from Sections 3 and 4 and report results in Figures 20 and 21 for Qwen-2.5 1.5B and 3B.

We observe that MMLU accuracy remains stable at approximately 61–62% and 64–65%, respectively for the two models, across all projection settings, indicating no degradation in general ability. This

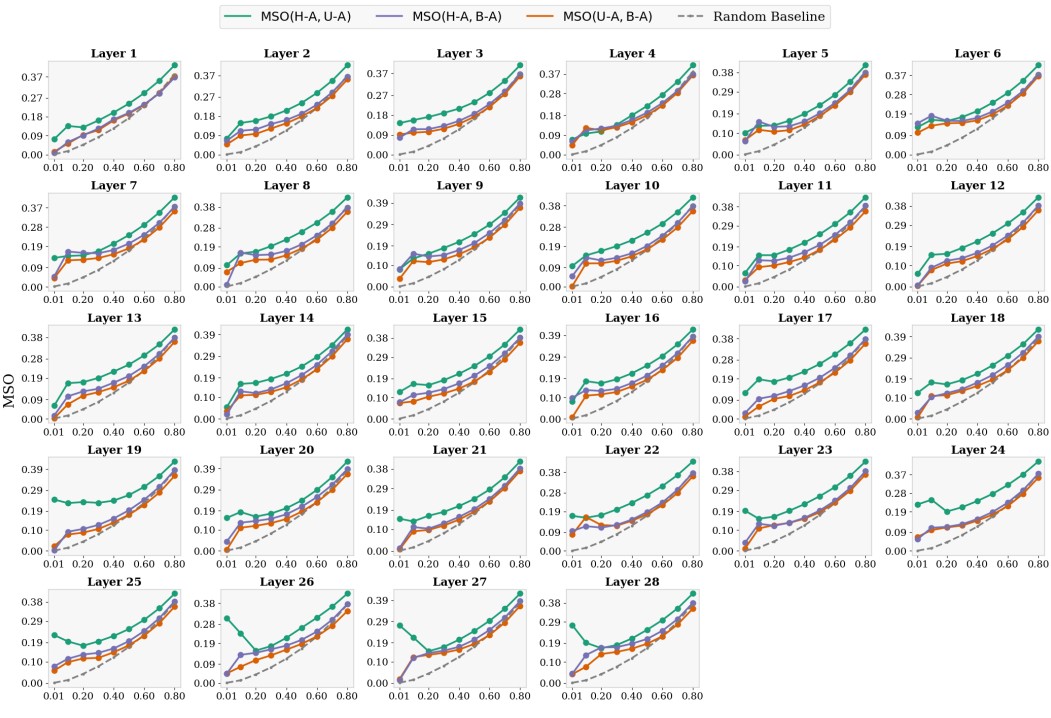

Figure 10: MSO at all layers for pairwise comparisons of the dominant weight subspaces from Harmful fine-tuned (H), Aligned (A), and Base (B) models (Qwen-2.5 1.5B).

strengthens our findings and confirms that our analysis is scientifically robust rather than an artifact of reduced model capability.

## J.2 RULING OUT OVER-REFUSAL AS A CONFOUNDING FACTOR

We test whether projection induces over-refusal, which would indicate that our safety results might be confounded by models incorrectly rejecting benign queries. To evaluate this, we measure refusal rates on benign prompts using XSTest (55), which ideally should not be refused. We repeat the experiments from Sections 3 and 4 and report results in Figures 20 and 21 for Qwen-2.5 1.5B and 3B.

Across all projection settings, refusal rates remain very low, around 3–4% and 5–6% respectively for the two models. These consistently low rates indicate that our safety metrics are robust and that the findings reported in the paper are not confounded by projection-induced over-refusal.

## K ADDITIONAL ACTIVATION SPACE ANALYSES

We do several experiments to complement the discussion in Section 6.

## K.1 ACTIVATION SPACE ANALYSIS ACROSS ALL LAYERS

We report activation-space results layer by layer to fully capture layer-specific behavior. These results are shown for Qwen-2.5 1.5B, in Figures 22, 23, and 24. We observe that our hypothesis and findings hold consistently across all layers.

## K.2 IMPACT OF USING EARLY TOKEN WINDOWS

We show the effect of using earlier token windows in Figure 25 for Qwen-2.5 1.5B and 3B. We observe that our results and hypothesis hold true in this setting as well.

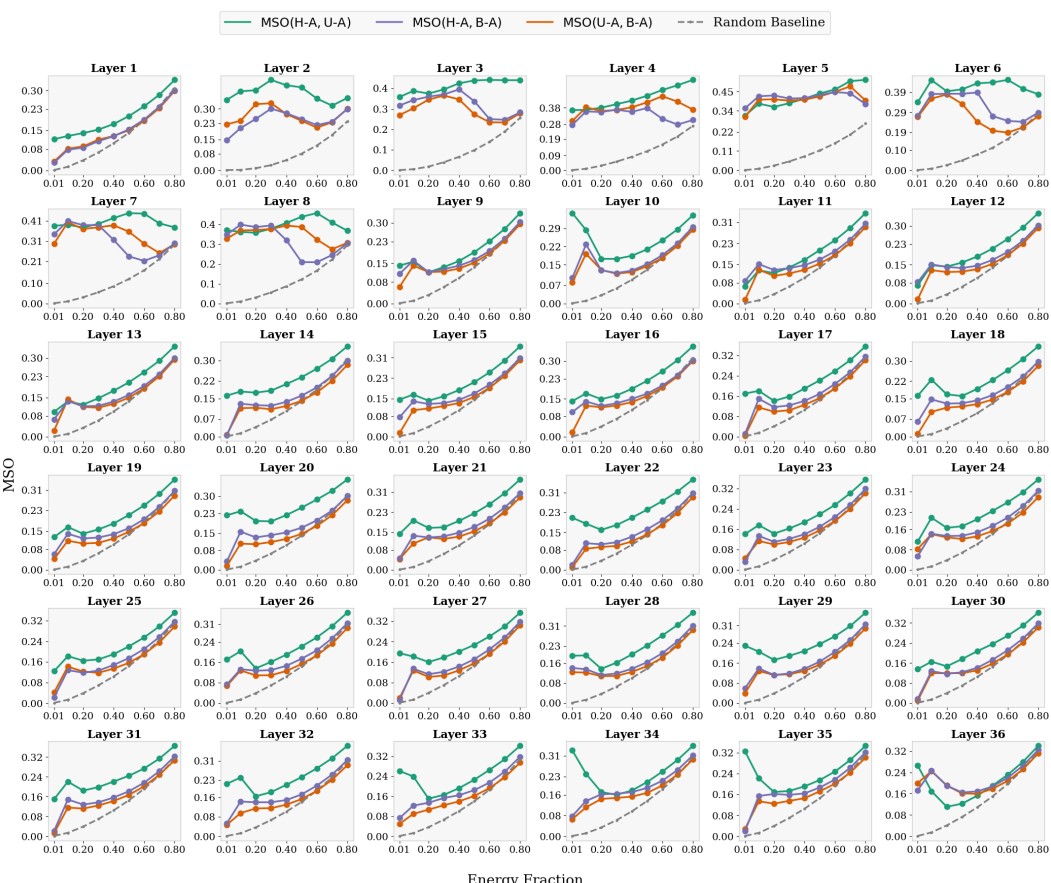

Figure 11: MSO at all layers for pairwise comparisons of the dominant weight subspaces from Harmful fine-tuned (H), Aligned (A), and Base (B) models (Qwen-2.5 3B).

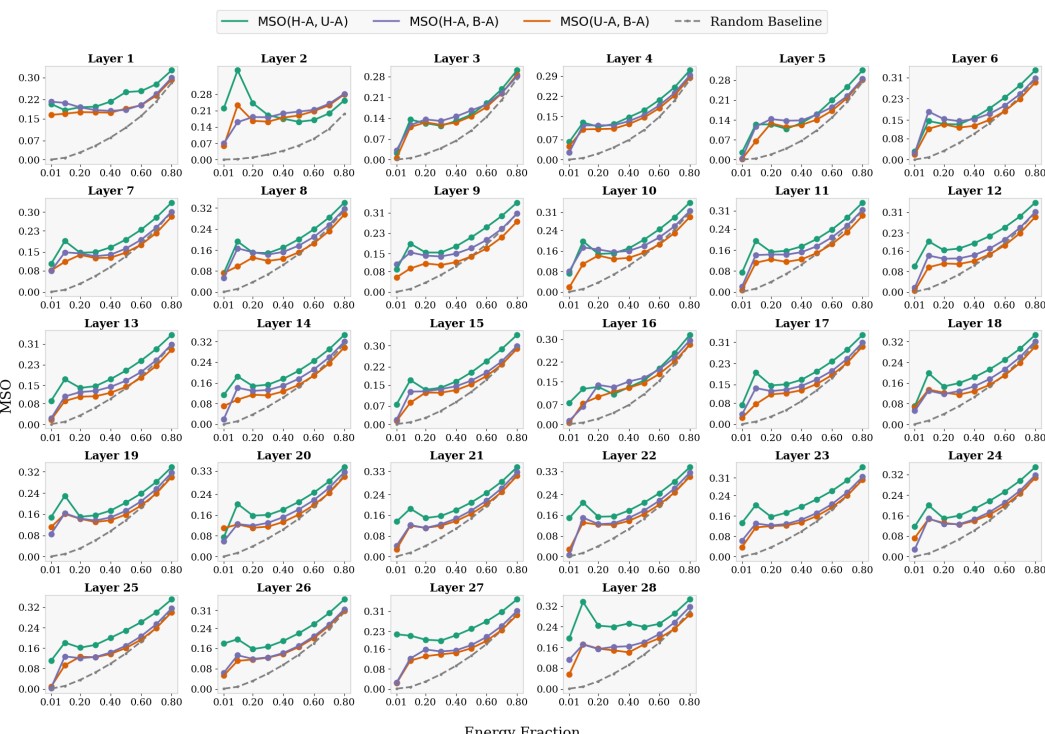

Figure 12: MSO at all layers for pairwise comparisons of the dominant weight subspaces from Harmful fine-tuned (H), Aligned (A), and Base (B) models (Qwen-2.5 7B).

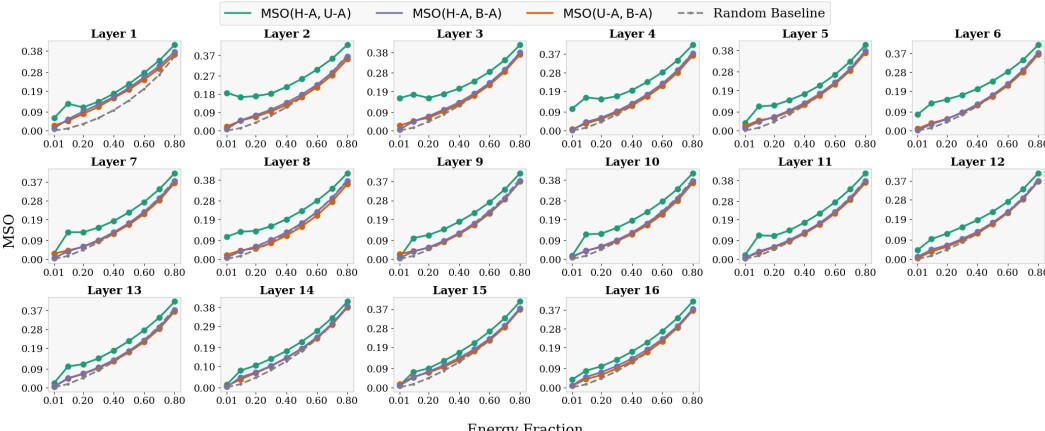

Figure 13: MSO at all layers for pairwise comparisons of the dominant weight subspaces from Harmful fine-tuned (H), Aligned (A), and Base (B) models (Llama-3.2 1B).

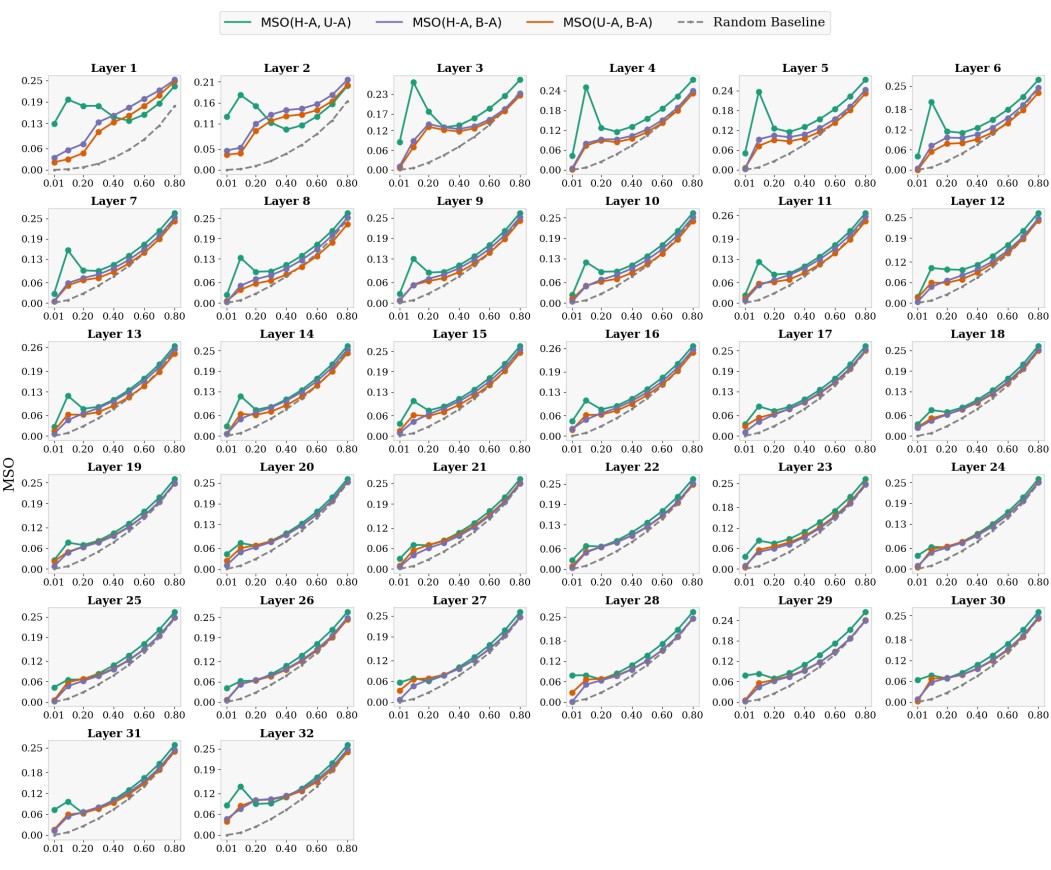

Figure 14: MSO at all layers for pairwise comparisons of the dominant weight subspaces from Harmful fine-tuned (H), Aligned (A), and Base (B) models (Llama-2 7B).

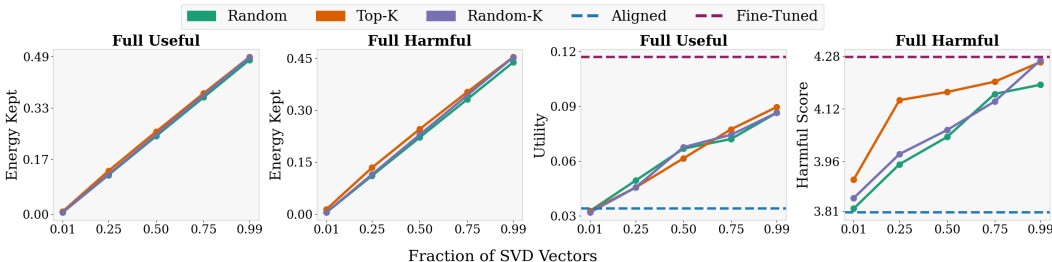

Figure 15: Parallel projection-based update schemes across varying SVD fractions (Llama-3.2 1B). We report the energy-kept ratio for models fine-tuned on Full Useful and Full Harmful data, utility for models fine-tuned on Full Useful, and harmfulness for models fine-tuned on Full Harmful. Model alignment was performed using DPO.

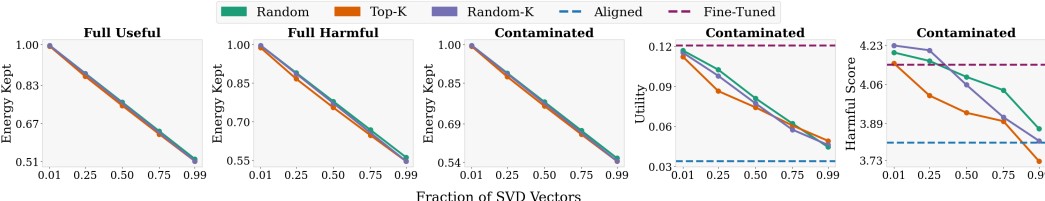

Figure 16: Orthogonal projection-based update schemes across varying SVD fractions (Llama-3.2 1B). We report the energy-kept ratio for models fine-tuned on Full Useful, Full Harmful and Contaminated data; and utility and harmfulness for models fine-tuned on Contaminated. Model alignment was performed using DPO.

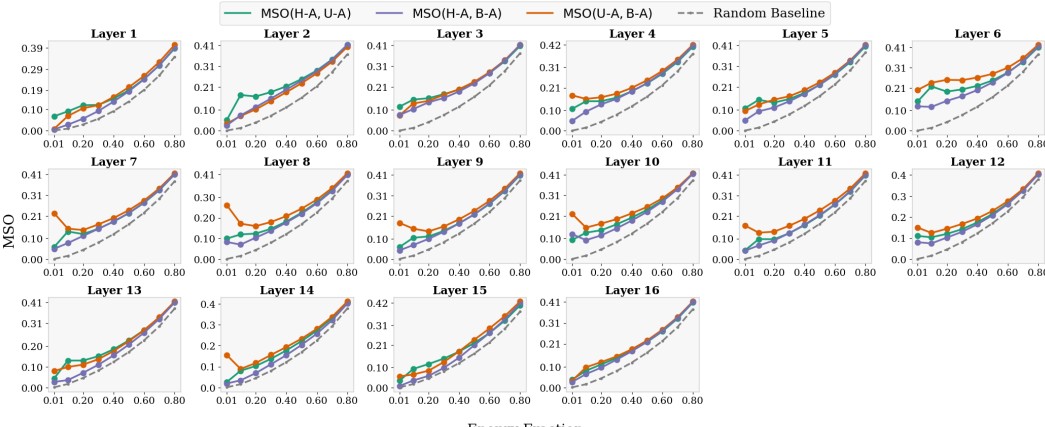

Figure 17: MSO at all layers for pairwise comparisons of the dominant subspaces from Harmful fine-tuned (H), Aligned (A), and Base (B) models (Llama-3.2 1B). Model alignment was performed using DPO.

### K.3 IMPACT OF USING REFUSAL PROMPTS

We investigate the effect of using a refusal-oriented dataset (harmless prompts from Extended Refusal, (57)) as the counterpart to harmful prompts. Instead of relying on a generic "useful" dataset such as MATH, this allows us to test whether our findings depend on the choice of benign data. As shown for Qwen-2.5 1.5B in Figure 26, the results remain consistent under this substitution.

## L    DISTINCTION FROM ACTIVATION STEERING METHODS

Our claims are compatible with the empirical success of activation steering methods. However, they target a stronger assumption that is not established by those works: namely, that there exists a relatively low-dimensional, stable "safety subspace" whose directions uniquely encode safety and can be edited in isolation, without affecting acquired utility.

Prior work shows that for a given model checkpoint and prompt distribution, one can reliably extract directions or small subspaces in the residual stream whose manipulation strongly affects refusal or a unique safety characteristic. Examples include single refusal directions (2), conditional activation steering (34), manifold steering for overthinking (26), and safety directions that become easier to find in higher-dimensional models (62). These results demonstrate local linear controllability: in a given model, at specific layers, there are directions along which safety behavior is highly sensitive.

What they do not by themselves establish is that these directions constitute a unique, model-level safety subspace that is (i) conceptually specific to "safety" rather than general behavior, and (ii) robust under fine-tuning or other parameter changes.

At the same time, a complementary body of work points toward a more entangled and multi-directional picture: safety and helpfulness often trade off along shared directions (62), and safety behavior can be mediated by structured sets of neurons or layers rather than a single feature (6; 78).

Taken together, these works suggest the following picture: activation steering can reliably surface high-leverage (but local) directions that influence refusal within a fixed model, but this does not imply that safety is represented by a stable linear subspace. Instead, the evidence is consistent with safety being encoded through distributed mechanisms that vary across tasks and checkpoints. Our claims target this stronger, subspace-level interpretation, rather than the more limited form of local controllability demonstrated by steering methods. This is particularly important because a growing body of work (e.g., SafeLoRA, SaLoRA, etc.) relies on this stronger assumption, despite there being no evidence that local controllability extends to global safety behavior.

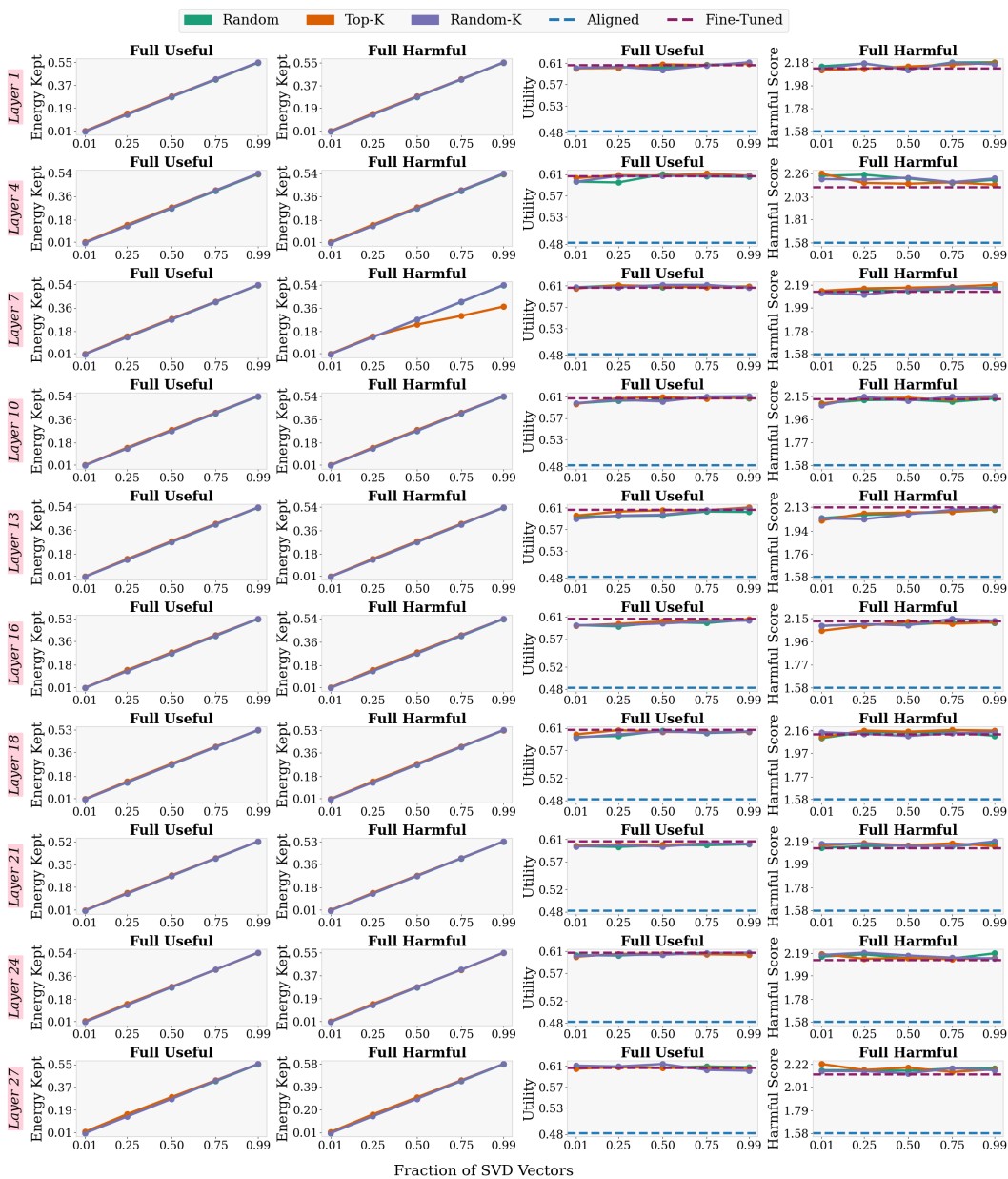

Figure 18: Parallel projection-based update schemes across varying SVD fractions (when applied to **a specific layer only**). We report the energy-kept ratio for models fine-tuned on Full Useful and Full Harmful data, utility for models fine-tuned on Full Useful, and harmfulness for models fine-tuned on Full Harmful.

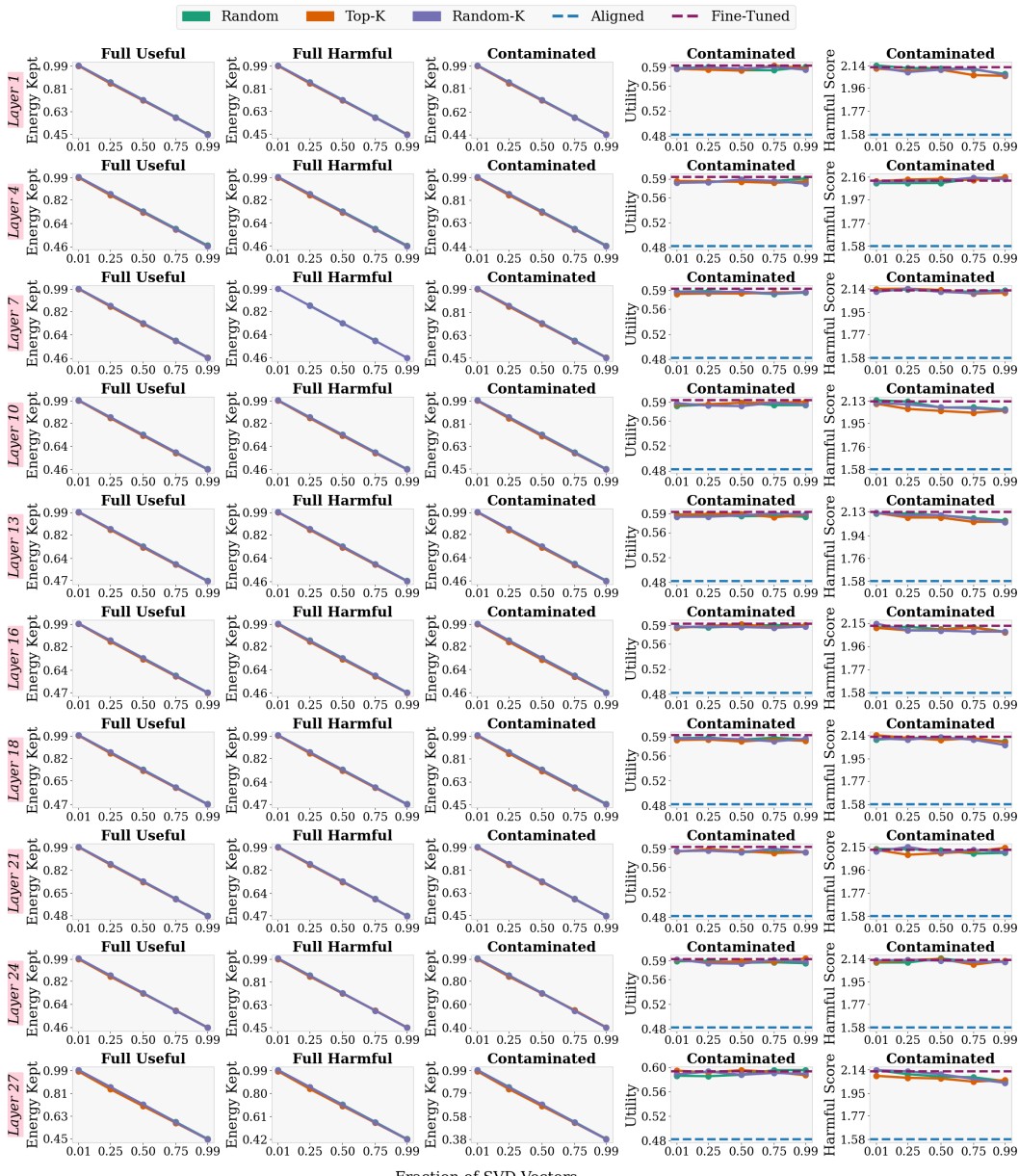

Figure 19: Orthogonal projection-based update schemes across varying SVD fractions (when applied to **a specific layer only**). We report the energy-kept ratio for models fine-tuned on Full Useful, Full Harmful and Contaminated data; and utility and harmfulness for models fine-tuned on Contaminated.

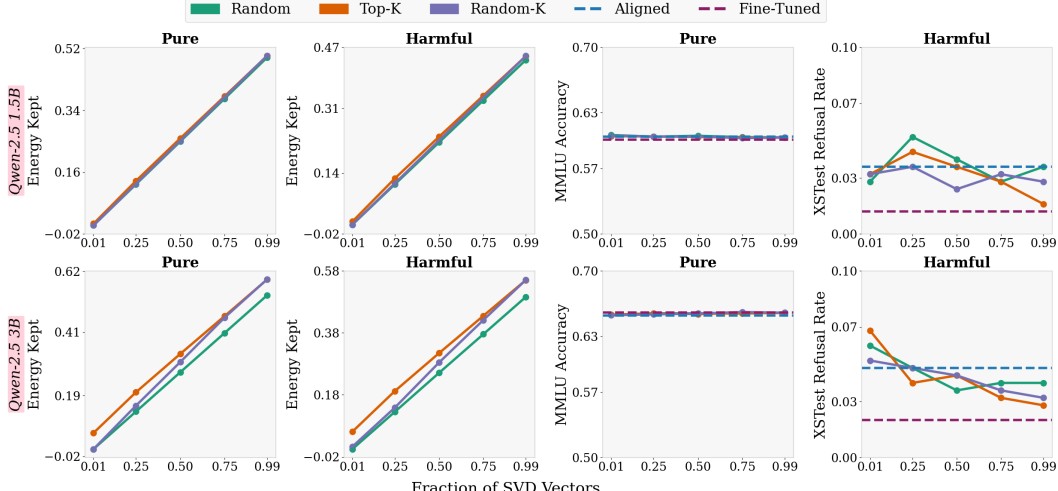

Figure 20: Parallel projection-based update schemes across varying SVD fractions. We report a general-ability metric (MMLU accuracy) for models fine-tuned on Full Useful, and refusal rates on benign prompts (XSTest) for models fine-tuned on Full Harmful.

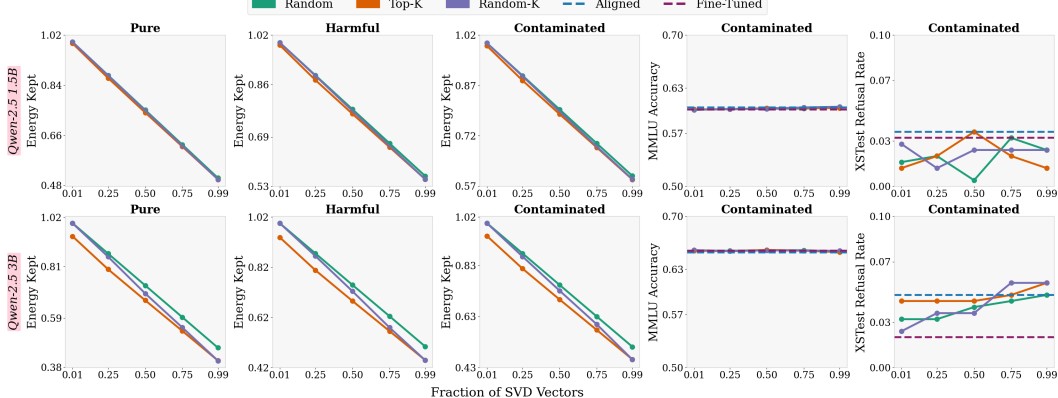

Figure 21: Orthogonal projection-based update schemes across varying SVD fractions. We report a general-ability metric (MMLU accuracy) and refusal rates on benign prompts (XSTest) for models fine-tuned on Contaminated.

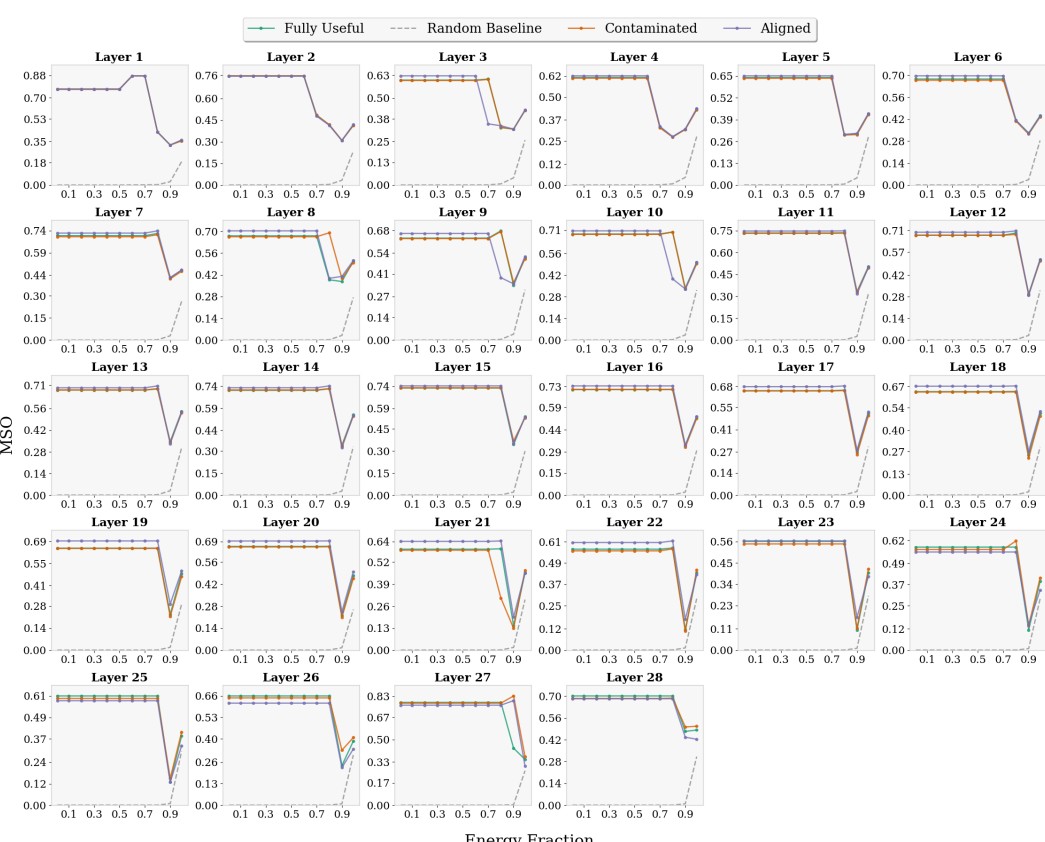

Figure 22: MSO across all layers for activations from Useful (Math) and Harmful (BeaverTails) prompt sets.

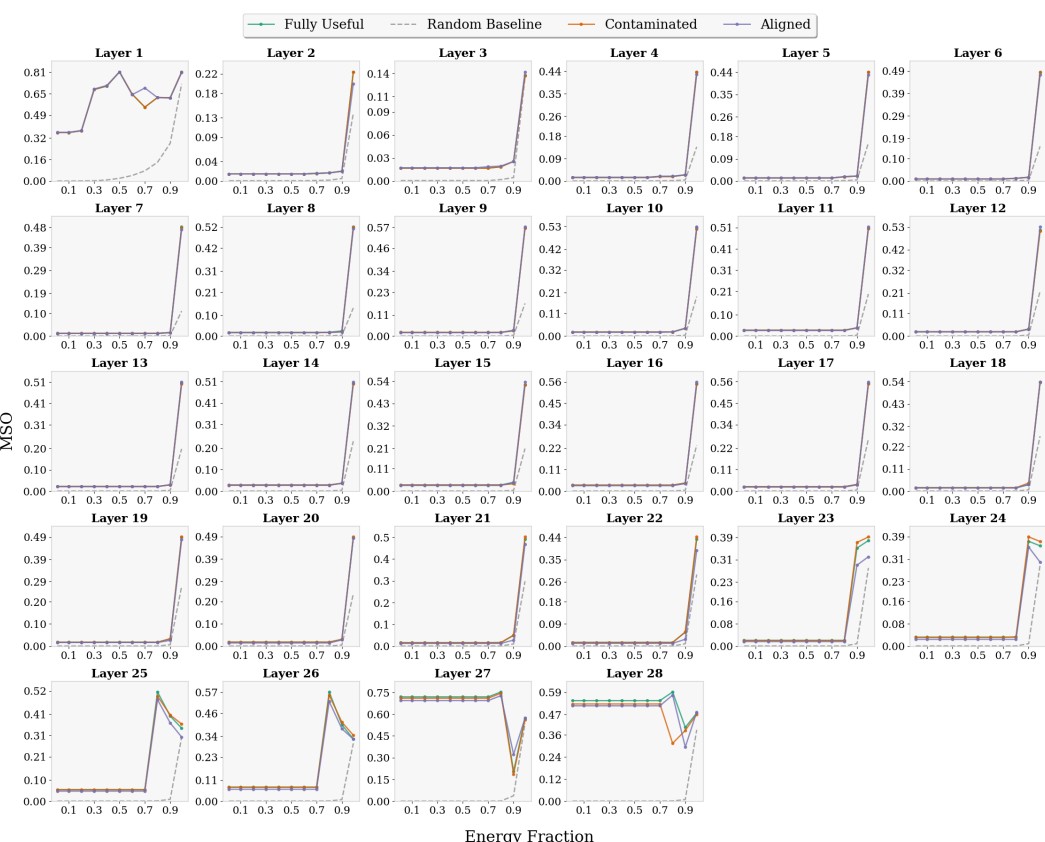

Figure 23: MSO across all layers for activations from Useful (Math) and Harmful (ToxiGen) prompt sets.

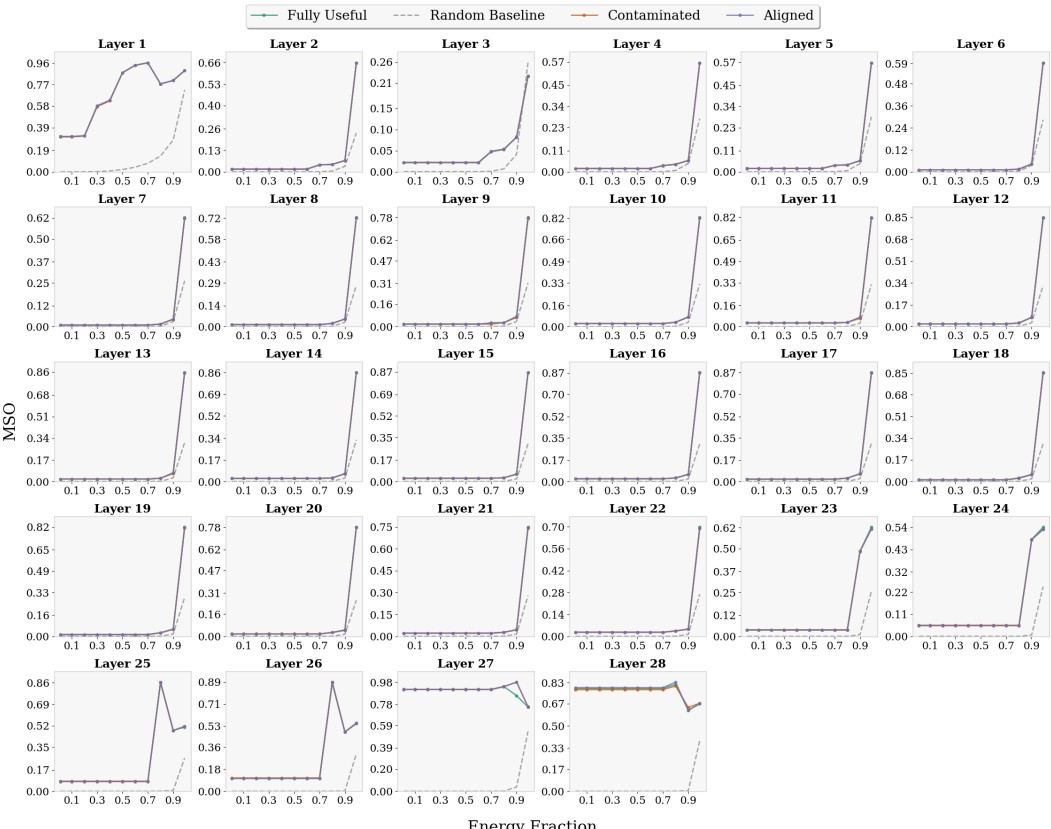

Figure 24: MSO across all layers for activations from different Harmful (BeaverTails, ToxiGen) prompt sets.

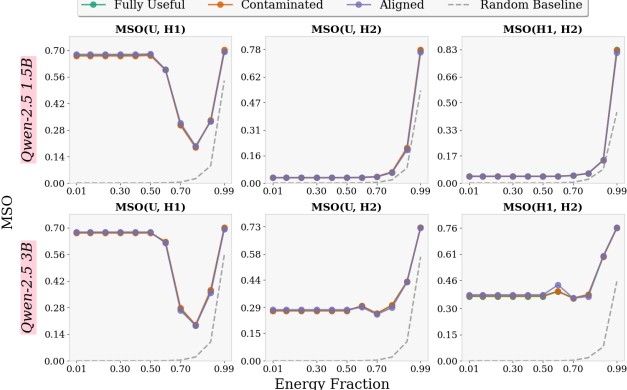

Figure 25: Average MSO (for early token windows) across layers in the 65–90% depth range for pairwise comparisons of activations from Useful (U) and multiple Harmful (H1, H2) prompt sets.

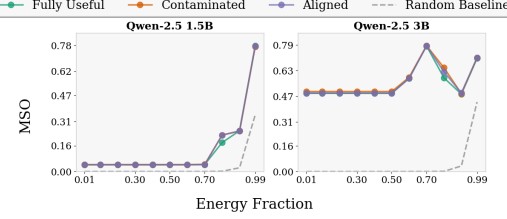

Figure 26: Average MSO across layers in the 65–90% depth range for activations from Refusal and Harmful prompt sets.

## M  DATASET DETAILS

We use the **MetaMathQA** dataset (75) for fine-tuning, which reformulates existing math problems from alternative perspectives without introducing new content. To evaluate performance, we rely on the **GSM8K** benchmark (9), a dataset of elementary-level math questions that require multi-step reasoning. Models are assessed based solely on the correctness of the final numerical answer. For our activation-based analysis, we sample prompts from the **MATH** dataset (19), which contains challenging, competition-style arithmetic problems.

**BeaverTails** (31) is a valuable dataset for studying safety by independently annotating question–answer pairs for both helpfulness and harmlessness. We use the training set to fine-tune models in both harmful and contaminated settings, and draw prompts from the test split for our activation-based experiments.

**AdvBench** (80) consists of 500 prompts designed to elicit a wide range of harmful behaviors, including profanity, threats, misinformation, discrimination, cybercrime, and other forms of dangerous or illegal content framed as instructions. We use this benchmark to quantify model harmfulness: higher success in responding to these prompts indicates greater unsafe behavior.

**ToxiGen** (15) is a large-scale dataset composed of both toxic and non-toxic statements. We use a subset of its prompts to analyze model activations in response to harmful content.

**MMLU** (18): The Massive Multitask Language Understanding (MMLU) benchmark evaluates general ability and knowledge. We use MMLU to test whether our harmfulness metric is confounded with general ability.

**Extended Refusal** (57) is a dataset of prompts curated to evaluate whether models refuse queries. In our work, we use its harmless subset as a refusal-oriented counterpart to harmful prompts.

**XSTest** (55) is a benchmark designed to measure refusal. We use XSTest to determine whether our projection methods induce over-refusal.

**PKU-SafeRLHF** (30) is safety-alignment dataset constructed for reinforcement learning with human feedback (RLHF), covering a wide spectrum of harmful and harmless interactions. We use PKU-SafeRLHF to safety-tune a model using DPO.

## N  LIMITATIONS

Our study focuses on linear subspaces, providing a principled first step toward understanding the geometric structure of safety alignment. While we do not explore non-linear representations, our framework could be extended in future work to capture richer geometric phenomena. Our experiments are restricted to open-weight models with publicly available base and aligned variants. These models provide a controlled and interpretable setting, though extending to production-level closed-source models remains an important direction for future work.

## O  USE OF LARGE LANGUAGE MODELS

Our use of LLMs is restricted to light writing assistance, including grammar polishing and enhancing clarity.

