# OpenReview forum: "Safety Subspaces are Not Linearly Distinct: A Fine-Tuning Case Study"
_ICLR.cc/2026/Conference — ICLR 2026 Poster_

### Official Review · Reviewer_M6zX · 2025-10-21

**Soundness:** 2
**Presentation:** 3
**Contribution:** 2
**Rating:** 4
**Confidence:** 4

**Summary:**

The paper tests whether “safety subspaces” exist as linear directions in weight or activation space that could preserve or filter safety during fine-tuning. The authors construct subspaces from SVD of alignment and safety deltas, then study how “useful” and “harmful” fine-tuning updates interact with these subspaces via parallel and orthogonal projections, and how their activation spaces overlap. Across five open-weight models, Top-K singular directions strongly affect both utility and harmfulness, orthogonal filtering reduces both together, and activation subspaces for useful and harmful prompts overlap. The paper concludes that distinct linear safety subspaces are unlikely and that projection-based defenses have limited value.

**Strengths:**

- **The question is important for alignment interpretability**. The paper asks whether safety can be isolated linearly in weight or activation space, which underlies many safety direction defenses. The experimental design addresses both weights and activations across several models.
- **Proper analyzing methods**. The projection schemes and subspace overlap measurement is theory-driven and well-formulated. The paper examine the so-called safety subspace from multiple aspects, which is solid.
- **The paper is generally clearly written**. I can easily follow the description and understand the claims.

**Weaknesses:**

My main concern of the paper is that the experiment settings in the paper are limited for the broad claims that "Safety subspace does not exist".

- **The training settings setup does not isolate safety signal**. The Safety tuning and downstream fine‑tuning use supervised fine‑tuning on text corpora with many properties besides safety. The "useful" task is MetaMathQA math problems and the "harmful" task uses BeaverTails unsafe data. This setup makes it hard to tell whether the top directions reflect safety or general learning, since both data sources induce large distribution shifts unrelated to safety per se. A stronger test would include a preference‑based safety objective that explicitly trains the model to discriminate safe versus unsafe generation (e.g. DPO [1]), avoiding introducing too much safety-agnostic signals.
- **Subspace extraction is unsupervised and layer‑agnostic**. Safety subspaces are extracted as the top‑k singular directions of layer-flattened weight updates. This can overweight non-safety layers and wash out safety features, as past works show that large weight shifts happen in later layers while safety mechanism might locate in mid layers [2]. I suggest the author to use layer-wise SVD. The author should also discuss supervised safety-subspace extraction (e.g. probing), which are shown to more effective in extracting safety-subspace. If none of these yield selective safety effects, the linear non‑separability claim is much more convincing.
- **The harmfulness metric is confounded with general ability**. Harmfulness is graded by a model judge on AdvBench, which may confound general ability (e.g. instruction following and writing quality) with harmfulness of the model. Figures 3 and Tables 2/5 show that orthogonal projection and even random projections reduce both utility and harmfulness together, which can be due to general degradation rather than safety improvement. I suggest the author also report more robust metrics like probability of toxicity generation or further examine if harmful score changes are caused by changes in general ability.
- **Interpretation of “energy kept” is unclear.** Energy-kept trends appear similar across choices, so it is hard to tell what semantic insight this quantity adds. More explanation will be helpful.

Therefore, the claims in this paper need narrower wording or stronger evidence:
- **Line 261: Alignment Directions Reflect General Learning, Not Safety**. The symmetry in Figure 2 and Table 1 shows that projecting useful and harmful updates onto top alignment directions increases both utility and harmfulness. This supports high‑impact general directions, but does not rule out safety linear-structure that is not captured by the SVD components. The paper should add the layerwise and supervised directions described above before generalizing this implication.
- **Line 340: No Selective Removal Is Possible**: Orthogonal projection removes top‑k directions and causes utility and harmfulness to drop together at similar rates (Figure 3; Tables 2 and 5). However, the top-k components is not relevant to safety anyway (as per the first experiment suggested). The result may reflect choice of subspace rather than a fundamental limit.
- **Line 461: Shared Subspaces Drive Behavior, Not Safety**: Section 6 compares activations from useful prompts (U) with two harmful sets (BeaverTails H1 and ToxiGen H2), then finds useful–harmful overlap often comparable to or exceeding harmful–harmful overlaps. Because H1 and H2 come from different sources and the activation analysis uses only the last token, the overlap patterns can reflect dataset and style similarities rather than safety.

From the experiments in the paper, what I can agree on is that **when we SFT models on some aligned / toxic dataset, the dominant subspace of the weights update contains not only safety-specific features, but also utility features.** While this negative result has its own contribution, we need stronger arguments to prove that safety is inherently entangled with general learning and are not linearly separable. The author should address above concerns before I will raise my score.

[1] Andrew Lee, Xiaoyan Bai, Itamar Pres, Martin Wattenberg, Jonathan K. Kummerfeld, and Rada Mihalcea. **A Mechanistic Understanding of Alignment Algorithms: A Case Study on DPO and Toxicity.** ICML, 2024.

[2] Rui Pan, Xiang Liu, Shizhe Diao, Renjie Pi, Jipeng Zhang, Chi Han, and Tong Zhang. **LISA: Layerwise Importance Sampling for Memory-Efficient Large Language Model Fine-Tuning.** NeurIPS, 2024.

**Questions:**

- Can you address my concern about the weaknesses?
- At line 242: "At the same time, while energy is evenly distributed, behavioral impact is not.". I don't quite get it. I see in Figure 2, all metrics are uniformly increasing with more fraction of SVD vector. How does this results implicate "behavioral impact is not evenly distributed"?
- Most results in the paper are observable, and the paper didn't give analysis on the counterintuitive results. Could you give some reasons on why your experiments shows different results than previous papers? - The implications in line:404 and line:461 are identical. Is it a typo?

---

> ### Author Response · Authors · 2025-11-20
>
> We thank the reviewer for the detailed review. We respond to each point below and provide multiple **new experiments** that further validate our claims.
>
> `W1: Choice of alignment method. “The training settings setup does not isolate safety signal. The Safety tuning and downstream fine‑tuning use supervised fine‑tuning ….”`
>
> Thank you for the suggestion.
>
> We now additionally perform alignment on a model using DPO. We then replicate all our weight-space analyses on the DPO-aligned model. The corresponding results are reported in **Appendix H, Figures 15-17**.
>
> Across these new experiments, we observe the same qualitative patterns as in the original setting: the high-energy directions of the DPO alignment updates do not form a safety-specific subspace, projecting task updates onto or off of these directions continues to jointly affect utility and harmfulness rather than isolating safety. Thus the entanglement we observe persists under a different alignment objective.
>
> This suggests that our conclusions are not an artifact of the particular alignment method originally used, but instead reflect a more inherent property of how safety and task-learning information is represented. Moreover, the maximum overlap observed in **Figure 17** is between a safety-specific DPO update and a utility-specific SFT update. This disentangles both safety and method choice as the causal reasons for the high overlap.
>
> `W2: “Subspace extraction is unsupervised and layer‑agnostic…” & W5: “Line 261: Alignment Directions Reflect General Learning, Not Safety...”`
>
> We would like to clarify the following: in our work, SVD is performed separately for each layer, not on a single layer-flattened tensor. For each layer, we compute SVD on the corresponding update ΔW, and all top-k singular directions, projections, and statistics are defined within that layer.
>
> To further examine where safety may reside in the network, we added two sets of explicitly layer-wise analyses. Note that all our projections were already applied per layer; here, we mean analyzing the effect of projecting **only a single layer in isolation**, or reporting the subspace overlap **for each layer separately**.
>
>
> **Per-layer overlap analysis (Appendix G, Figures 10-14)**: We complete the analysis of Section 5 by reporting per-layer MSO results. Across all models and layers, the strongest overlap is never between alignment/safety updates and harmful updates, which is not what one would expect if safety were represented as a shared linear component concentrated in particular layers.
>
> **Layer-localized projection experiments (Appendix I, Figures 18-19)**: We ask whether safety and learning decouple at any single layer by repeating the projection experiments from Sections 3 and 4, but restricting projections to individual layers (while keeping other layer updates as is). Projecting out directions at a single layer produces almost no discernible change in behavior (even when varying k), suggesting that meaningful behavioral transformations require coordinated changes across many layers. We project every third layer (10 layers total), and our conclusion remains consistent across the network: no individual layer exhibits a clean separation between safety and learning directions, and projecting safety-aligned directions at a single layer does not selectively suppress harmful behavior while preserving utility.
>
> Together, these results indicate that no layer contains a linearly separable safety subspace, and we do not see the kind of layer-local principal directions one would expect if safety mechanisms were cleanly isolated in mid layers.
>
>
>
>
> `W3: Harmfulness evaluation may be confounded with general ability degradation: “The harmfulness metric is confounded with general ability....”`
>
> Thank you for the suggestion!
>
> We agree that a judge-based harmfulness score could be partially confounded with general ability (e.g., instruction following, coherence), and we explicitly test for this.
>
> To do so, we measure general capability using MMLU (a widely-used common general understanding benchmark). We repeat the projection experiments from Sections 3 and 4 and report MMLU results in **Figures 20 and 21 (Appendix J.1)**. Across all projection settings, MMLU accuracy remains unchanged with projection (>60% for all models), while the harmfulness metric still changes.
>
> This stability of MMLU demonstrates that the projected models do not suffer a systematic degradation in general ability, even though harmfulness (as measured on AdvBench by a model judge) decreases. In other words, the safety effects we report are not simply a byproduct of globally “breaking” the model or impairing its overall competence, which addresses the concern that our harmfulness metric might be confounded with general capability.

---

> ### Author Response · Authors · 2025-11-20
>
> `W4: “Interpretation of “energy kept” is unclear....” & Q2: “At line 242: "At the same time, while energy is evenly distributed, behavioral impact is not."...?”`
>
> Energy kept ratio is a purely geometric quantity which tells us how much of the update we are retaining (in norm), but not how that retained part affects behavior. We use this as a control variable. In particular, we show that useful and harmful updates place similar amounts of energy into the alignment/safety subspaces (Figures 2-3).
>
> The sentence “energy is evenly distributed, but behavioral impact is not” is about the difference between how much update energy is kept and what that energy does. In Figure 2, for any fixed energy-kept level (that is, for a fixed fraction of the update norm retained), we can compare the top-k and random projections. The key observation being that, while the energy-kept curves are similar, the behavioral effects are not: Top-k projections cause much larger changes in utility and harmfulness than random projections that keep a comparable amount of energy. In other words, the update energy is broadly spread across many directions, but the behaviorally important part of that energy is highly concentrated in a small set of high-impact directions; which directions we choose matters much more than how much norm we keep.
>
> Finally, we emphasize that we use the Top-k singular directions not because we assume they are safety-specific, but because they are empirically the most behaviorally influential directions. In later experiments, we also sweep the subspace dimension all the way up to full rank and analyze overlaps of the full update subspaces. Across all subspace sizes, including the full space, we do not see a regime where a selectively safety-only linear region appears.
>
> `W6: “Line 340: No Selective Removal Is Possible…”`
>
> Our second experiment addresses a different question than the first, and is specifically designed to test the core assumption behind subspace-based defenses.
>
> In the first experiment, we study the effect of projecting updates along the “safety” directions for purely useful and purely harmful fine-tuning. This shows that the top-k directions derived from alignment/safety updates are not in themselves safety-selective. However, it is critical to note that they are most effective for safety.
>
> Injecting harmful data during fine-tuning is a common strategy for intentionally misaligning models. In the second experiment, we instead look at contaminated fine-tuning updates that mix useful and harmful signals, and we project these updates onto subspaces orthogonal to the candidate safety subspaces. This setup directly corresponds to the standard subspace-defense intuition: if safety information lives in a special subspace, then removing those components should selectively gate the harmful part while preserving the useful part.
>
> What we observe is that, with change in projection, utility and harmfulness change together at similar rates. In other words, for these candidate safety subspaces, there is no regime where harmful behavior can be removed without paying a comparable cost in usefulness. This is exactly the failure mode we highlight: even when we move to the “orthogonal complement” setting that subspace defenses rely on, we do not see evidence of safety and utility separating linearly.

---

> ### Author Response · Authors · 2025-11-20
>
> `W7: Activation overlap may be style-driven; last-token only analysis. “Section 6 compares activations from useful prompts …”`
>
> ### Last token analysis
>
> We have added multiple new experiments to address this. In **Appendix K.2 (Figure 25)**, we now report activation-overlap results for earlier token windows and in **Appendix K.1 (Figures 22-24)**, we report layer-wise residual-stream overlaps. Across token positions and layers, we observe the same pattern: activation subspaces for harmful and benign prompts remain highly overlapped, and we do not find a safety-selective subspace.
>
> ### Potential confound from different dataset choices
>
> Across token positions and layers, we consistently find that the MSO between useful and harmful (U and H1/H2) prompts is higher than the MSO between the two harmful datasets (H1 and H2), even though H1 and H2 are closer in task to each other than either is to U. If dataset source were the dominant factor, we would instead expect H1-H2 to show the highest overlap. The fact that cross-task (useful-harmful) overlap exceeds within-task (harmful-harmful) overlap shows that our conclusions are robust to dataset choice.
>
> To further probe the role of the benign dataset choice, we replace MATH with a refusal-oriented benign dataset: harmless prompts from Extended Refusal, used as the counterpart to harmful prompts. This controls for the possibility that MATH behaves differently from safety-adjacent benign data. As shown in **Appendix K.3 (Figure 26)**, the qualitative overlap structure remains unchanged under this substitution: harmful and benign prompts still occupy highly overlapping activation subspaces, and no safety-selective subspace emerges.
>
> Taken together, these results suggest that the observed activation-space entanglement is not driven primarily by last-token effects or by dataset differences. Instead, they support our interpretation that harmful and benign prompts share common representational subspaces throughout the network.
>
>
>
>
> `Q3: “Most results in the paper are observable, and the paper didn't give analysis on the counterintuitive results. Could you give some reasons on why your experiments shows different results than previous papers?”`
>
> Prior work (e.g., SafeLoRA, SaLoRA) mostly looks at local, task-specific fixes. These approaches appear to work because they’re tested in settings where safety happens to degrade more slowly than performance improves, making the trade-off look manageable. But as our experiments show, this behavior is unreliable and doesn’t hold up across different tasks, models, or fine-tuning conditions.
>
> Our results instead focus on global, high-energy, linear SVD-style subspaces built from full-model updates. When we analyze these, we see a very different story: the supposed “safety” directions are brittle and unstable, and are highly correlated with general acquired utility. Rather than revealing a clean, robust safety subspace, the findings demonstrate the difficulty in isolating safety as a single, stable linear factor that can be edited reliably.
>
> `Q3: The implications in line:404 and line:461 are identical. Is it a typo?`
>
> Thanks for pointing this out. We have corrected it in the updated version of the paper (Lines 406 and 465).
>
> ---
>
> We genuinely appreciate the reviewer’s comments, which have helped us improve our paper. If our rebuttal resolves the concerns highlighted, we request you to **kindly consider increasing your score**. We are happy to answer any remaining questions you might have!

---

> > ### Comment · Reviewer_M6zX · 2025-11-25
> >
> > Thank you for the additional experiments. My main concerns have been resolved by the new evidence you provided. Although the findings are mostly negative, the paper contributes to a more nuanced understanding of safety subspaces. I therefore support acceptance of this paper.

---

> ### Author Response · Authors · 2025-11-25
>
> Thank you! We truly appreciate your detailed, constructive reviews, which helped us strengthen the claims in our paper and refine the validity of our negative results. We’re glad our rebuttal addressed your concerns, leading you to recommend acceptance and raise your score.
>
> Studying where plausible and promising assumptions in AI safety hold, and where they do not, is essential for building more reliable and trustworthy AI systems. We are glad that our work contributes to that discussion. Thank you again for your support.

---

### Official Review · Reviewer_QX9G · 2025-10-24

**Soundness:** 3
**Presentation:** 3
**Contribution:** 3
**Rating:** 6
**Confidence:** 3

**Summary:**

The paper examines whether linear subspaces derived from alignment and safety deltas can selectively control safety, and whether activation patterns for harmful prompts occupy distinct regions. The main observations are that Top-K SVD directions are strong general-impact directions for both utility and harmfulness, orthogonal filtering trades off both together, and activation subspaces of useful and harmful prompts overlap at many layers. These results argue that simple linear projection defenses have limited selectivity.

**Strengths:**

- Evaluations span multiple Llama and Qwen variants and include analyses under contamination. This breadth supports external validity of the main pattern.
- The paper explains the operators, datasets, and scoring choices clearly enough that others can reproduce or extend the tests.
- Demonstrating limits of global SVD-style filtering is actionable for researchers who might otherwise rely on it as a safety control.

**Weaknesses:**

- Broad-claim relative to evidence. The experiments show that global Top-K SVD directions are not safety-selective. That does not exclude the existence of linear safety structure discoverable through other methods, such as multiple independent refusal directions or concept cones[1]. The claim should be scoped accordingly.
- Activation analysis focuses on last-token states, but refusal dynamics often appear early in the response[2] instead in the last token. I would like to see whether using earlier token windows or residual-stream features changes the overlap picture.
- Useful and harmful prompts differ in source and style. Some of the observed overlap could reflect stylistic differences.

[1] Tom Wollschläger et, al. *The Geometry of Refusal in Large Language Models: Concept Cones and Representational Independence. 2025.
[2] Andy Arditi et, al. _Refusal in Language Models Is Mediated by a Single Direction._ 2024

**Questions:**

- The paper introduces both the general alignment update ($\Delta_A$) and a specific safety-tuning update ($\Delta_S$), but the analysis often groups them. What are the difference between the subspaces derived from $\Delta_A$ and $\Delta_S$?

---

> ### Author Response · Authors · 2025-11-20
>
> Thank you for your detailed review. Below, we address all points raised and include **multiple new experiments** to strengthen our claims.
>
> `W1: Scope of claim. “Broad-claim relative to evidence. The experiments show that global Top-K SVD directions are not safety-selective. That does not exclude the existence of linear safety….”`
>
> We agree that our conclusions do not definitively disprove the existence of safety subspaces. Rather, our results show that there is substantial evidence supporting a general-ability view of model updates, and little to no evidence supporting the alternative hypothesis of a stable, linearly separable safety subspace. We now clarify this scope in the paper.
>
> Our experiments go beyond showing that global Top-K SVD directions are not safety-selective: (i)The first two sets of experiments use the top-k singular vectors of alignment/safety updates as candidate safety subspaces and test whether projecting onto/away from them isolates safety from utility. (ii) The later experiments look for any preferential geometric alignment between safety-related updates, harmful updates, and capability updates. Across models, layers, and energy thresholds, we consistently find that overlap between safety-specific deltas and harmful/task deltas have a lower similarity lower than cross-task overlaps (e.g., useful-harmful), which suggests that safety subspaces are not linearly separable.
>
> This is strong (though not formal) evidence that no global linear safety subspace exists in weight space.  We have scoped the claim accordingly: we mention that we find no evidence that any linear subspace, whether in weight or activation space, captures safety-specific behavior in isolation, even though we cannot disprove its existence. We have now revised our claims in lines 75, 87, and 93 in the updated paper.
>
>
> `W2: “Activation analysis focuses on last-token states, but refusal dynamics often appear early in the response[2] instead in the last token. I would like to see whether using earlier token windows or residual-stream features changes the overlap picture.”`
>
> Thank you for the suggestion.
>
> We have added the following experiments to address this concern.
>
> **Early token window activation analysis**: **Appendix K.2 (Figure 25)** shows activation-overlap results using intermediate tokens, instead of just using the last token. The qualitative pattern is unchanged: activation subspaces for harmful and benign prompts remain highly overlapped, and no safety-selective subspace emerges when focusing on earlier token windows.
>
> **Layer-by-layer activation analysis**: **Appendix K.1 (Figures 22-24)** shows results across all activations from all layers. Again, we observe the same behavior: harmful and benign prompts occupy strongly overlapping subspaces at each layer, with no layer that exhibits a clean, linearly separable safety-specific subspace.
>
>
> `W3: “Useful and harmful prompts differ in source and style. Some of the observed overlap could reflect stylistic differences.”`
>
> We agree that, in principle, differences in source and style between useful and harmful prompts could be a confound. However, our results suggest that in our case it is not a confound but it makes our conclusions more robust.
>
> First, across all layers and token positions, activation subspaces for harmful and useful prompts remain highly overlapped. More importantly, we find that the MSO between harmful and useful activations is higher than the MSO between two harmful datasets, even though the latter are closer in task. If prompt style were the main factor, we would expect the two harmful datasets (which are more similar to each other than to MATH) to show the highest overlap. Instead, we observe the opposite pattern: cross-task (useful-harmful) overlap is stronger than within-task (harmful-harmful) overlap.
>
> Second, to directly probe the role of benign dataset choice and style, we replace MATH with a refusal-oriented benign dataset: harmless prompts from Extended Refusal as the counterpart to harmful prompts. This controls for the possibility that MATH, as a generic capability dataset, might behave differently from safety-adjacent benign data. As shown in **Figure 26 (Appendix K.3)**, the qualitative overlap structure remains unchanged under this substitution: harmful and benign prompts still occupy highly overlapping activation subspaces, and no safety-selective subspace emerges.
>
> Taken together, these observations indicate that prompt style is unlikely to be a meaningful confound. The fact that (i) cross-task overlaps exceed within-task overlaps in the harmful datasets, and (ii) the results are stable when replacing MATH with a refusal-style benign dataset, reinforces our main conclusion about entanglement rather than weakening it.

---

> > ### Author Response · Authors · 2025-11-20
> >
> > `Q1: “The paper introduces both the general alignment update (ΔA) and a specific safety-tuning update (ΔS), but the analysis often groups them. What are the difference between the subspaces derived from ΔA and ΔS?”`
> >
> > We distinguish between the two updates as follows. ΔA is the general alignment update obtained from a standard instruction-/preference-tuning pipeline; it improves helpfulness, refusals, instruction-following, and general response quality. As such, it is the closest proxy to real-world “production” alignment, but it inevitably mixes safety signals with non-safety signals (formatting, style, task-following, etc.). To more cleanly isolate safety, we also study ΔS, which is trained purely on safety data and is explicitly targeted at harmlessness behavior.
> >
> > In our analyses, we derive candidate subspaces separately from ΔA and ΔS (via SVD of each update) and use both as projection bases. Empirically, the subspaces induced by ΔA and ΔS behave very similarly: in both cases, top-k directions act as high-impact, task-agnostic learning directions rather than safety-selective ones. Projecting task updates onto or away from the ΔS subspace does not reveal a cleaner separation between safety and capability than projecting with ΔA, and we do not observe a regime in which ΔS yields a distinct “safety axis” that ΔA misses.
> >
> > Intuitively, ΔS can be viewed as the safety-focused component of alignment, while ΔA contains ΔS-like safety corrections plus additional non-safety refinements. The fact that neither ΔA-derived nor ΔS-derived subspaces produce a linearly separable safety direction supports our interpretation that post-training alignment updates, whether broad or explicitly safety-only, operate within the same general-purpose learning subspaces, rather than carving out safety-unique directions.
> >
> >
> >
> > ---
> >
> > We thank the reviewer again for the helpful suggestions, which have helped us strengthen our claims. If our responses adequately address your concerns, we request you to **kindly consider increasing your score**. We are happy to clarify any remaining points you might have!

---

> > > ### Author Response · Authors · 2025-11-26
> > >
> > > This is a gentle reminder regarding our rebuttal. We know the review process can be demanding and apologize if you were already intending to revisit this review soon. We truly appreciate the attention you have given our submission, and we hope our replies have resolved your concerns. Thank you once more for such a careful and insightful review, which helped us refine our claims. Please feel free to reach out with any remaining questions!

---

### Official Review · Reviewer_Xrmm · 2025-10-29

**Soundness:** 2
**Presentation:** 3
**Contribution:** 3
**Rating:** 4
**Confidence:** 4

**Summary:**

This work challenges the prior assumption that safety relevant features lie in a specific subspace of the weight space, hence, can be isolated and defended against jailbreaks. Using 2 metrics (Energy-kept ratio, Mode Subspace Overalp), authors find that safety subspaces are entangled with general utility subspaces hindering the applicability of the prior assumption and defense mechanisms.

**Strengths:**

1. **Significance:** The paper addresses broadly believed hypothesis that a concept-related features are essentially contained in a specific lower dimensional subspace.
2. **Novelty:** While mostly building upon [1], the authors take a critical standpoint to better investigate the source of the positive results in [1] and tradeoffs it poses (utility drops similarly with the removal of "harmful subspace"). There have been works to observe that most jailbreak defenses are also likely to hurt the general utility of the model, yet a dedicated investigation was lacking in the literature.
3. **Writing:** The writing is mostly clear and easy to follow.

**Weaknesses:**

Although the paper claims their findings for both weight space and activation space, the activation space seems to be relatively underexplored. Considering that LLM steering is predominantly done on its activations (Activation Addition, Directional Ablation, Manifold Steering, Angular Steering etc.), I think the paper does not deliver sufficiently in this important aspect. The following questions are to specify why exactly I think the current study is insufficient.

**Questions:**

1. If there is no such concept subspaces like "safety" in activation space, how would authors explain the success of activation steering methods?
2. Do results of this paper have any implications on linear separability hypothesis [2] which led many to believe there exist linear concept subspaces [3, 4]?
3. What is the reason to choose the MATH dataset as a counterpart for "harmful prompts"? Would using any refusal dataset instead of a generic "useful" dataset impact the observation?

___

Overall, I believe the topic of the paper is relevant and important for the ICLR community, and the paper offers nontrivial contribution (see Strength). Therefore, I am willing to increase my score upon satisfactory responses to the questions raised.

___

### References

1. Chia-Yi Hsu, Yu-Lin Tsai, Chih-Hsun Lin, Pin-Yu Chen, Chia-Mu Yu, and Chun-Ying Huang.Safe lora: the silver lining of reducing safety risks when fine-tuning large language models, 2025.
2. Kiho Park, Yo Joong Choe, Victor Veitch. The Linear Representation Hypothesis in Language Models. NeurIPS 2023
3. Yao Huang, Huanran Chen, Shouwei Ruan, Yichi Zhang, Xingxing Wei, Yinpeng Dong. Mitigating Overthinking in Large Reasoning Models via Manifold Steering. arXiv:2505.22411
4. Rachel S.Y. Teo, Laziz U. Abdullaev, Tan M. Nguyen. The Blessing and Curse of Dimensionality in Safety Alignment. COLM 2025.

---

> ### Author Response · Authors · 2025-11-20
>
> We appreciate the reviewer’s comments and feedback. In the following, we clarify all raised concerns and questions in detail.
>
> `W1 & Q1: Safety spaces and activation steering. W1: “Although the paper claims their findings for both weight space and activation space, the activation space seems to be relatively underexplored….” Q1: “If there is no such concept subspaces like "safety" in activation space, how would authors explain the success of activation steering methods?”`
>
> Our claims are compatible with the empirical success of activation steering methods. However, they target a stronger assumption that is not established by those works: namely, that there exists a relatively low-dimensional, stable “safety subspace” whose directions uniquely encode safety and can be edited in isolation, without affecting acquired utility.
>
> Prior work shows that for a given model checkpoint and prompt distribution, one can reliably extract directions or small subspaces in the residual stream whose manipulation strongly affects refusal or a unique safety characteristic. Examples include single refusal directions [Arditi et al., 2024], conditional activation steering [Lee et al., 2025], manifold steering for overthinking [Huang et al., 2025], and safety directions that become easier to find in higher-dimensional models [Teo et al., 2025]. These results demonstrate local linear controllability: in a given model, at specific layers, there are directions along which safety behavior is highly sensitive.
>
> What they do not by themselves establish is that these directions constitute a unique, model-level safety subspace that is (i) conceptually specific to “safety” rather than general behavior, and (ii) robust under fine-tuning or other parameter changes.
>
> At the same time, a complementary body of work points toward a more entangled and multi-directional picture: safety and helpfulness often trade off along shared directions [Teo et al., 2025], and safety behavior can be mediated by structured sets of neurons or layers rather than a single feature [Zhao et al., 2025; Chen et al., 2024].
>
> Taken together, these works suggest the following picture: activation steering can reliably surface high-leverage (but local) directions that influence refusal within a fixed model, but this does not imply that safety is represented by a stable linear subspace. Instead, the evidence is consistent with safety being encoded through distributed mechanisms that vary across tasks and checkpoints. Our claims target this stronger, subspace-level interpretation, rather than the more limited form of local controllability demonstrated by steering methods. This is particularly important because a growing body of work (e.g., SafeLoRA, SaLoRA, etc.) relies on this stronger assumption, despite there being no evidence that local controllability extends to global safety behavior.
>
> `Q2: “Do results of this paper have any implications on linear separability hypothesis [2] which led many to believe there exist linear concept subspaces [3, 4]?”`
>
> The Linear Representation Hypothesis (LRH) [Park et al., 2023] and related work are often interpreted as suggesting that high-level concepts correspond to 1-D or low-rank linear features in activation space. We see our conclusions as refining this view rather than contradicting its empirical core. LRH and follow-up studies convincingly show that many properties are linearly decodable in a fixed model, and that some behaviors can be modulated by perturbing specific directions.
>
> However, more recent analyses already indicate that this linear picture is incomplete for complex concepts: Park et al. [2023] identify inherently multi-dimensional and even circular features, Teo et al. [2025] and Pan et al. [2025] find multiple, partially overlapping safety directions rather than a single safety axis, and manifold-based approaches (Yousefpour et al. [2025]) argue that concepts often occupy distributed manifolds rather than isolated 1-D directions.
>
> Our claims are aligned with this more nuanced perspective. We do not dispute that “safe vs. unsafe” behavior is linearly decodable to some degree in a given checkpoint, nor that linear steering directions can be constructed. The point is that such linear decodability does not automatically imply the existence of a stable, linearly separable concept subspace that is uniquely responsible for safety and survives changes to the model.
>
> Instead, safety appears to be an emergent property of more general-purpose representational subspaces that also encode capability, with linear probes and steering directions acting as approximate readouts or control knobs rather than revealing a single, canonical “safety feature.” In this sense, our conclusions are consistent with the current trajectory of LRH-style work: linear structure is real and useful, but complex concepts, especially safety, are better thought of as entangled, multi-dimensional representations than as a single global direction.

---

> ### Author Response · Authors · 2025-11-20
>
> `Q3: “What is the reason to choose the MATH dataset as a counterpart for "harmful prompts"? Would using any refusal dataset instead of a generic "useful" dataset impact the observation?”`
>
> We chose MATH as the main benign/useful counterpart to harmful prompts for two reasons. (i) It provides a clean capability axis. MATH primarily probes arithmetic and reasoning ability, which is naturally distinct from refusal or safety behavior. This makes it well suited for studying whether safety-related directions can be geometrically separated from pure capability updates. (ii) It avoids conflating safety and capability in the “useful” task. Using a benign-refusal dataset would blur the distinction between capability and safety signals in the deltas we analyze, making it harder to interpret whether any observed overlap is due to genuine entanglement or simply because both sides are partially safety-oriented.
>
> To check that our conclusions do not depend on this specific choice, we also ran analyses using Extended Refusal (a refusal-oriented dataset) on harmless prompts in place of MATH. We observe the same qualitative overlap structure: harmless and harmful prompts still occupy highly overlapping subspaces, and no clean safety-only subspace emerges (**Appendix K.3, Figure 26**).
>
>
> Thus, this shows that MATH serves as a clean and interpretable capability baseline, and the additional experiments with Extended Refusal indicate that our conclusions are robust to replacing MATH with a refusal-style benign dataset.
>
>
> ---
>
> ## References
>
> - Arditi, A. et al. *Refusal in Language Models is Mediated by a Single Direction.* 2024.
> - Chen, J. et al. *Towards Understanding Safety Alignment: A Mechanistic Perspective from Safety Neurons.* arXiv:2406.14144, 2024.
> - Huang, Y. et al. *Mitigating Overthinking in Large Reasoning Models via Manifold Steering.* arXiv:2505.22411, 2025.
> - Lee, B. W. et al. *Programming Refusal with Conditional Activation Steering.* ICLR 2025.
> - Lermen, S. et al. *LoRA Fine-Tuning Efficiently Undoes Safety Training in Llama 2-Chat 70B.* 2024.
> - Park, K., Choe, Y. J., and Veitch, V. *The Linear Representation Hypothesis in Language Models.* NeurIPS 2023.
> - Qi, X. et al. *Fine-Tuning Aligned Language Models Compromises Safety, Even When Users Do Not Intend To!* 2023.
> - Yousefpour, A. et al. *Representation Bending for Large Language Model Safety*. ACL 2025.
> - Teo, R. S. Y. et al. *The Blessing and Curse of Dimensionality in Safety Alignment.* COLM 2025.
> - Zhao, Y. et al. *Understanding and Enhancing Safety Mechanisms of LLMs via Safety-Specific Neuron.* ICLR 2025.
> ---
>
> Thank you again for the thoughtful feedback. If our rebuttal adequately addresses your concerns, we **kindly ask you to consider raising your score**. We are happy to clarify any remaining questions you might have!

---

> > ### Comment · Reviewer_Xrmm · 2025-11-20
> > **Concerns mostly addressed**
> >
> > I thank the authors for their thorough responses. The comparative discussion with prior works in the rebuttal clears most of the ambiguities I experienced while reading the manuscript. Thus, I am increasing my score to 6 upon my promise, and hope that authors can update the main text accordingly.

---

> > > ### Author Response · Authors · 2025-11-24
> > >
> > > Thank you! We’re glad our rebuttal helped clarify your concerns and gave you confidence in raising your score. We will make sure to incorporate the discussion into the final version of the paper. Thanks again for the detailed feedback and thoughtful scientific discussions.

---

> ### Author Response · Authors · 2025-12-02
>
> Thanks again for the suggestions and for **raising your score**! We have added the comparative discussion with activation steering methods in Appendix L.

---

### Official Review · Reviewer_YiuK · 2025-11-01

**Soundness:** 3
**Presentation:** 4
**Contribution:** 2
**Rating:** 4
**Confidence:** 3

**Summary:**

The paper investigates whether safety behaviors in large language models correspond to distinct linear subspaces that can be isolated without degrading general capability. It constructs “safety subspaces” from the principal components of fine-tuning updates between different stages of fine-tuning, then measures their geometric overlap using metrics such as Mutual Subspace Overlap and Energy Kept Ratio. Across multiple model families, the authors find that safety and utility update directions share most of their variance and projection along these subspaces reduces both helpfulness and harmlessness simultaneously. The results suggest that safety alignment is not linearly separable but deeply entangled within the same representational space as capabilities.

**Strengths:**

1. The paper formalizes “safety subspaces” concretely by building them from principal components of weight differences. Although PCA on weight deltas is common in task-vector or model diffing work, applying it here provides an interesting way to identify safety regions beyond analyses that focus only on activations popular in recent AI safety works.

2. It introduces two sound measures of subspace overlap that are intuitively meaningful, providing a clear alternative to relying solely on downstream evaluation metrics to quantify model diffing methods.

3. The presentation is clear: each research question is addressed in a dedicated section and supported by plots and analysis and the claims in this paper is well supported. The reported metrics and ablations are convincing, and the additional validation in activation space increases confidence that the findings are not artifacts of a single parameterization.

**Weaknesses:**

1. The conclusions may depend on the specific design choices for subspace dimension and PCA aggregation (it seems that SafeMERGE supports a form of layer-wise separability). The paper fixes the number of principal components and the layer aggregation strategy without sensitivity analysis, so the observed overlap might partially reflect those implementation choices rather than a conclusive property.

2. The experiments use a single alignment method and do not test how geometric overlap changes under different alignment objectives, especially under common alignment methods like RLHF and DPO, which usually yield better alignment. Without this comparison, it is unclear whether the observed entanglement is inherent to model structure or specific to the alignment method used and prevailing in all safety representations.

3. The evaluation measures helpfulness and harmlessness separately but does not analyze their joint behavior on adversarial inputs. Testing whether projection affects over-refusal, for example by using XSTest for benign prompts that should not be refused, would better assess whether safety information can be isolated.

**Questions:**

1. Would you say your conclusions align with the findings of Wei et al. (2024) despite the methodological differences? From my understanding, they appear to have reached a similar conclusion.

2. Do you expect the observed entanglement behavior to change under Mixture-of-Experts (MoE) model families, where safety and capability routing may be distributed across distinct experts?

3. How is the k in the top-k principal components chosen, and what proportion of the total variance does this selection typically explain?


Wei, B., Huang, K., Huang, Y., Xie, T., Qi, X., Xia, M., Mittal, P., Wang, M., Henderson, P. (2024). Assessing the Brittleness of Safety Alignment via Pruning and Low-Rank Modifications. arXiv:2402.05162. https://arxiv.org/abs/2402.05162

**Details Of Ethics Concerns:**

Safety related

---

> ### Author Response · Authors · 2025-11-20
>
> Thank you for the detailed review. Below, we systematically address all issues raised and conduct **multiple new experiments** to strengthen our claims.
>
> `W1: Subspace dimensions and layer-wise separability. “The conclusions may depend on the specific design choices for subspace dimension and PCA aggregation….”`
>
> We would like to clarify that throughout all our experiments, we vary k (the number of subspace components considered). Across this entire range, from very small fractions to using the full vector space, we consistently find that the identified directions are not specific to safety. The first two experiments (Sections 3, 4) analyze the top principal components (with k varied widely), while the third (Section 5) finds no evidence that any choice of subspace selection would make isolation possible. Taken together, all our experiments show that no choice of subspace meaningfully isolates safety.
>
> To address the concern about layer-wise aggregation (and the possibility of layer-wise separability), we have added explicit layer-wise analyses via two new sets of experiments:
>
> **Per-layer overlap analysis (Appendix G, Figures 10-14)**: We now report per-layer MSO results for all models completing the analysis in Section 5. At every layer and for all models, the strongest overlap is never between alignment and harmful updates; indicating that safety is not captured by a single shared linear component. Instead, cross-task overlaps (e.g., useful-harmful) are consistently larger. This pattern is stable across layers and architectures.
>
>
> **Layer-localized projection experiments (Appendix I, Figures 18-19)**: We explicitly ask whether safety and learning decouple at any single layer by repeating the projection experiments from Sections 3 and 4, but restricting projections to individual layers. A notable finding from these experiments is that layer-wise projections are not only agnostic to safety and utility, but they are also largely ineffective overall. Projecting out directions at a single layer produces almost no discernible change in behavior (even when varying k), suggesting that meaningful behavioral transformations require coordinated changes across many layers. We project every third layer (10 layers total), and our conclusion remains consistent across the network: no individual layer exhibits a clean separation between safety and learning directions, and projecting safety-aligned directions at a single layer does not selectively suppress harmful behavior while preserving utility.
>
>
> Thus, our findings are robust to a layer-wise aggregation strategy, and they argue against the existence of a linearly separable safety subspace even under more fine-grained, layer-local analyses.
>
>
>
> `W2: Choice of alignment method. “The experiments use a single alignment method and do not test how geometric overlap changes under different alignment objectives, especially under common alignment methods like RLHF and DPO..”`
>
> Thank you for the suggestion.
>
> We agree that it is important to test whether our conclusions are robust to the choice of alignment objective.
>
> We now additionally perform alignment on a model using DPO. We then replicate all our weight-space analyses on the DPO-aligned model. The corresponding results are reported in **Appendix H, Figures 15-17**.
>
> Across these new experiments, **we observe the same qualitative patterns as in the original setting**: the high-energy directions of the DPO alignment updates do not form a safety-specific subspace, projecting task updates onto or off of these directions continues to jointly affect utility and harmfulness rather than isolating safety. Thus, the entanglement we observe persists under a different alignment objective.
>
> This suggests that our conclusions are not an artifact of the particular alignment method originally used, but instead reflect a more inherent property of how safety and task-learning information is represented. Moreover, the maximum overlap observed in **Figure 17** is between a safety-specific DPO update and a utility-specific SFT update. This disentangles both safety and method choice as the causal reasons for the high overlap.

---

> ### Author Response · Authors · 2025-11-20
>
> `W3: Ruling out over-refusal as a confound. “The evaluation …Testing whether projection affects over-refusal, for example by using XSTest for benign prompts that should not be refused, would better assess whether safety information can be isolated.”`
>
> Thank you for the suggestion!
>
> We have added a test for checking whether projection induces over-refusal, which would indicate that our safety results might be confounded by models incorrectly rejecting benign queries. For this, we evaluate whether projection increases refusals on benign prompts using XSTest.
>
> We repeat the projection experiments from Sections 3 and 4 and, in addition to our original usefulness/harmlessness metrics, measure refusal rates on XSTest for Qwen2.5-1.5B and Qwen2.5-3B. The results, reported in **Appendix J.2 (Figures 20-21)**, show that refusal rates remain consistently low across all projection settings, around 3-4% for Qwen2.5-1.5B and 5-6% for Qwen2.5-3B. These consistently low rates indicate that our safety metrics are robust and that the findings reported in the paper are not confounded by projection-induced over-refusal.
>
> `Q1: “Would you say your conclusions align with the findings of Wei et al. (2024) despite the methodological differences? From my understanding, they appear to have reached a similar conclusion.”`
>
> Yes; broadly speaking, our conclusions are consistent with those of Wei et al. (2024), despite the methodological differences. Wei et al. probe brittleness primarily via pruning and low-rank removal, while we study geometric structure via SVD-based update analysis, MSO, and activation-space overlap. Both lines of work support a similar high-level picture: safety alignment can be implemented through relatively low-rank behavioral modifications, but these do not manifest as a clean, linearly separable “safety subspace” that can be isolated without affecting capabilities. We view our results as complementary to their findings.
>
>
> `Q2: “Do you expect the observed entanglement behavior to change under Mixture-of-Experts (MoE) model families, where safety and capability routing may be distributed across distinct experts?”`
>
> We did not run MoE-specific experiments, so we cannot make a strong empirical claim here, but we expect our core conclusions to carry over. In typical MoE architectures, expert routing is learned during pretraining and is driven by generic capability and distributional structure in the data, rather than by downstream safety objectives. To our knowledge, there is currently no evidence that safety behaviors (e.g., refusals) are cleanly routed to distinct safety experts, and safety alignment is usually applied post hoc via relatively small updates (e.g., RLHF/DPO-style preference tuning) that are unlikely to substantially reorganize the routing policy or carve out dedicated safety experts. Even if some degree of specialization emerged (e.g., certain experts being more active on harmful prompts), our results suggest that within activated experts, safety and task-learning directions are still geometrically entangled rather than linearly separable. This is an interesting setting where our framework could be extended and could be part of future work.
>
> `Q3: “How is the k in the top-k principal components chosen, and what proportion of the total variance does this selection typically explain?”`
>
> We already vary k across its full range in all of our experiments, so our conclusions do not depend on a particular choice of subspace dimensionality.
>
> In Experiments 1 and 2 (Sections 3 and 4), we sweep the number of top singular directions from very small k up to the full vector space. In Experiments 3 and 4 (Sections 5 and 6), we rely on MSO that compares the relative geometry of high-energy modes over a range of energy thresholds rather than fixing a single k.
>
> Across all of these settings, the qualitative trends are invariant to the choice of k, which suggests that the observed entanglement is not an artifact of how we select the subspace dimensionality. Specifically, all our figures sweep k over its full range (visible on the x-axis of each plot). For typical models, the top-k components explain a substantial fraction of update energy; with the behavioral trends (joint effects on utility and harmfulness) remaining unchanged as k increases. We also report full-rank settings, which yield the same conclusions. Taken together, these results clarify our procedure for selecting the projection dimension: we sweep over a range of k with our conclusions being consistent across different values.
>
> ---
>
> We sincerely appreciate your thoughtful feedback, which has helped us refine and strengthen our claims. If our rebuttal resolves the concerns you raised, we request you to **kindly consider increasing your score**. We are happy to address any additional questions you might have!

---

> > ### Author Response · Authors · 2025-11-26
> >
> > This is a gentle reminder regarding our rebuttal. We fully understand the reviewers’ workload and apologize if you were already planning to look at it soon. Your time and care in reviewing our work mean a lot to us, and we hope our responses have addressed your concerns thoroughly. Thank you again for the thoughtful review, which played an important role in strengthening our claims. If anything remains unclear, we would be glad to clarify!

---

### Official Review · Reviewer_hV1p · 2025-11-01

**Soundness:** 3
**Presentation:** 3
**Contribution:** 3
**Rating:** 6
**Confidence:** 4

**Summary:**

This paper empirically investigates whether safety subspaces exist in LLM weight or activation space that uniquely encode alignment through 4 studies: Projecting helpful and harmful updates into alignment/safety subspaces, orthogonally filtering contaminated updates, comparing subspace overlaps among useful/harmful/alignment directions, and analyzing activations for harmful vs. benign prompts. The authors find that high-impact directions amplify both utility and harmfulness, and activations for harmful and benign prompts substantially overlap. The results indicate safety is entangled with general learning rather than linearly separable, so projection or filtering along “safety” directions cannot selectively suppress harmful behavior without similar utility loss; this challenges subspace-based defenses and raise a need for alternative strategies for maintaining safety during continued training.

**Strengths:**

- The paper does an extensive empirical analysis extends from weight update directions to activations, and evaluates across multiple open source LLM families (Llama, Qwen) and fine tuning regimes (helpful, harmful, contaminated).
- The work draws a concrete practical implication: subspace-based defenses cannot selectively suppress harmful behavior without proportional utility loss, guiding future safety strategies away from linear subspace filtering.
- The results provide consistent evidence in both weight and activation spaces that harmful and benign signals substantially overlap, indicating these directions are not clearly separable.

**Weaknesses:**

- The focus of this work focus on defense mechanisms based on linear orthogonal projections, leaving non-linear alternatives unexplored.
- The paper reports activation-space results as averages over a mid-depth slice (about 65–90% of layers) rather than layer by layer, which may not fully capture layer-specific behavior.
- The paper lacks a theoretical investigation of the results, relying on empirical evidence without a formal framework explaining why safety and utility directions overlap. This is reasonable for an empirical study, but a concise theoretical model would strengthen the contribution.

**Questions:**

- Can you provide layer-by-layer activation-space results to better demonstrate the layer-specific behavior.?
- Is there a theoretical explanation or intuition for why high impact directions amplify both helpfulness and harmfulness, and under what conditions might they be separable? For example, would adding a contrastive objective during training help disentangle directions associated with helpfulness and harmfulness?
- Can we learn a system prompt that, when used for conditioning, reduces the overlap between safety and utility directions, potentially making subspace based defenses practical even though current finetuning results do not show clear separability?

---

> ### Author Response · Authors · 2025-11-20
>
> We thank the reviewer for the thoughtful feedback. Below we address each point raised, and conduct **new experiments** to strengthen our claims.
>
> `W1: “The focus of this work focus on defense mechanisms based on linear orthogonal projections, leaving non-linear alternatives unexplored.”`
>
> This comment highlights an important point and we agree with the reviewer that the scope of our work is restricted to linear projection-based defenses. Thus, our empirical results do not directly extend to non-linear alternatives and we acknowledge this limitation in the paper.
>
> This restriction is deliberate: one of our goals is to study the effectiveness and limitations of recent defenses (SafeLoRA, SaLoRA, SafeMerge, etc.) that assume safety-relevant updates introduced during fine-tuning can be linearly separated in weight and activation space.
>
> Exploring whether safety is non-linearly separable, and how non-linear methods could be used to exploit such structure, is an interesting and challenging direction that we consider out of scope for this work but plan to pursue in future work. We also welcome suggestions and further discussion on promising ways to approach this problem.
>
> `W2 & Q1: Activation-space results over other layers. W2: “The paper reports … which may not fully capture layer-specific behavior.”
> Q1: “Can you provide layer-by-layer activation-space results to better demonstrate the layer-specific behavior?”`
>
> Thank you for the helpful suggestion!
>
> We now include layer-by-layer activation-space MSO curves for all layers (see **Appendix K.1 and Figures 22, 23, 24**) in the revised manuscript. These results show that, across the entire depth of the model, harmful and benign prompts maintain consistently high subspace overlap, and no individual layer exhibits safety-specific separability. This confirms that our conclusions are not an artifact of averaging over mid-deep layers: the absence of a linearly separable “safety subspace” in activation space holds uniformly across layers.
>
> `W3 and Q2: Theoretical intuition behind results. W3: “The paper lacks …. This is reasonable for an empirical study, but a concise theoretical model would strengthen the contribution.” Q2: “Is there a theoretical explanation or intuition for why high impact directions amplify both helpfulness and harmfulness, …?”`
>
> We conjecture that high-impact directions amplify both helpfulness and harmfulness because they are precisely the directions along which the model is most sensitive, as determined during pretraining. Any fine-tuning objective; whether helpful or harmful, tends to exploit these directions, so both types of updates get “channeled” through the same subspace and are amplified similarly.
>
> Moreover, the most influential subspaces in a model are established during pretraining, where the model learns the core structure of language and most of its general capabilities. Any subsequent post-training or fine-tuning primarily adjusts or reinforces specific behaviors on top of these pretrained foundations. What makes alignment (and safety) distinct is that it is applied entirely post-hoc, rather than being built into the model during pretraining.
>
> We hypothesize that this is a key reason why safety becomes so deeply confounded with general learning directions. Because safety is not embedded in the model’s pretrained weights, but instead added afterward through alignment procedures, it does not occupy a clean or isolated subspace. Instead, safety signals end up intertwined with the broader directions that shape the model’s general behavior.
>
> In principle, separability could arise if training explicitly regularized the geometry of updates associated with helpful vs. harmful behavior. For example: one could penalize the inner product (or subspace overlap) between directions associated with helpful and harmful updates in weight or activation space, or directly encourage orthogonality between subspaces spanned by helpful vs. harmful gradients
>
> However, we emphasize that standard contrastive objectives primarily act on metric separation of representations (making harmful vs. benign activations far apart) rather than on subspace geometry (making the corresponding update directions orthogonal). Two behaviors can be far apart in representation space yet still lie in the same 1D or low-dimensional subspace. Thus, a naive contrastive objective on activations does not guarantee linearly separable update directions, especially in weight space where pretrained capabilities must be preserved.
>
> Our results suggest that achieving genuine subspace-level disentanglement would require strong, explicitly geometric regularization that competes with the pretrained model’s preferred high-impact directions, raising nontrivial trade-offs with capability. Characterizing when such objectives can enforce separability without significantly degrading performance is an interesting and important theoretical question.

---

> ### Author Response · Authors · 2025-11-20
>
> `Q3: “Can we learn a system prompt that, when used for conditioning, reduces the overlap between safety and utility directions, …?”`
>
> Our analysis primarily focuses on the geometry of updates in weight space and activation space. System prompts act only at inference time: they modify the activations conditioned on the input, but they do not change how fine-tuning updates are distributed across weight-space directions. Prompting influences behavior, and it can reshape the activation geometry for a given input, potentially changing which directions are most active. In principle, it could be possible to optimize a system prompt to reduce activation-space overlap between helpful and harmful prompts according to a metric like MSO.
>
> However, this would be a challenging high-dimensional optimization problem with non-trivial solutions as it is not obvious that a single fixed prompt can simultaneously preserve utility and enforce low overlap.
>
> ---
>
> Thank you again for the constructive suggestions - they have improved the clarity of our claims. If our responses adequately address your concerns, **we kindly ask you to consider increasing your score**. We are happy to clarify any further concerns you might have!

---

> ### Author Response · Authors · 2025-11-26
>
> This is a gentle reminder regarding our rebuttal. We fully understand the reviewers’ workload and apologize if you were already planning to look at it soon. We truly appreciate the time you have already spent reviewing our work. We hope our responses have addressed your concerns as thoroughly as possible. Thank you again for the careful and thoughtful review, which genuinely helped us strengthen our claims. We are happy to clarify any remaining questions you might have!

---

> > ### Comment · Reviewer_hV1p · 2025-11-26
> > **Response to authors rebuttal**
> >
> > I thank the authors for their detailed response and agree that the rebuttal has adequately addressed my concerns. I therefore decide to maintain my positive score.

---

> ### Author Response · Authors · 2025-11-26
>
> Thank you. We’re glad our rebuttal adequately addresses all your concerns. Thanks again for your detailed review.

---

### Comment · Area_Chair_HhbZ · 2025-11-23
**Next Steps Following Authors’ Rebuttal: Review Rebuttal and Participate in Discussion**

Dear Reviewers,

Thank you very much for your thoughtful evaluations of this paper.

Now that the authors have submitted their rebuttal, I kindly ask you to take the following steps (if you have not done so already):

- Read the other reviews as well as the authors’ response.
- Consider whether the rebuttal and additional comments affect your assessment of the paper.
- Engage in interactive discussion with the authors **before November 25**, encouraging a dynamic exchange rather than a one-sided rebuttal.

The current reviews for this paper are mixed. Your contributions at this stage are essential for forming a well-informed final decision. I therefore ask that you reassess your views in light of the authors’ responses and the broader discussion among reviewers.

I am happy to join and support the discussions between you and the authors. Please feel free to share your thoughts and participate actively in the discussion.

Thank you once again for your service to ICLR 2026.

Best regards,

 AC

---

### Author Response · Authors · 2025-12-02
**Summary of discussion period (avg. score increased to 6 from 4.8, despite two reviewers not participating)**

This is our attempt to summarize the discussion period as openly and honestly as we can.
All of the points below can be verified directly from the discussion record.

---

`Reviewers M6zX and Xrmm` both stated in their **original reviews (Official Review)** that they would be **willing to raise their scores if their concerns were addressed**.


`Reviewer M6zX` wrote that “*My main concerns have been resolved by the new evidence you provided.*” They **raised their score from 4 to 8** and added, “*I therefore support acceptance of this paper.*”

`Reviewer Xrmm` mentioned that “*the rebuttal clears most of the ambiguities.*” They raised their **score from 4 to 6**, noting “*Thus, I am increasing my score to 6 upon my promise.*”

Since the above reviewers stated in their **original reviews** that they would raise their scores if their concerns were addressed, this **removes any possible doubt that collusion** could have occurred and influenced their score updates.

`Reviewers YiuK and QX9G` were **not able to participate** in the discussion before it ended (with original scores 4 and 6, respectively). We addressed each of their concerns meaningfully, and believe they would have increased their score if they had participated in the discussion.

`Reviewer hV1p` stated that the “*rebuttal has adequately addressed my concerns,*” giving a score of 6.

These updates **improved our average to 6 from 4.8 (without doubt of any collusion)** at the time the discussion period ended abruptly, **even without the participation** of reviewers YiuK and QX9G.

---

We hope this context makes clear that our detailed rebuttal addressed the key concerns of all reviewers in a thorough and satisfactory way. We request that the AC consider these factors when evaluating our paper in these unexpected scenarios. We remain grateful for the time and effort going into evaluating our work.

---

> ### Author Response · Authors · 2025-12-02
> **Summary of major experiments and revisions**
>
> Below is a summary of all the major experiments and revisions we incorporated in our paper during the discussion period in response to reviewer feedback.
>
> ---
>
> `Effect of using DPO as an alignment method (Reviewers YiuK, M6zX)`
>
> Repeating our weight-space analyses on a DPO-aligned model yields the same qualitative patterns (**Appendix H, Figures 15-17**) as in our original experiments. This confirms that our observations hold under all alignment methods.
>
> `Ruling out the possibility of layer-wise separation (Reviewers YiuK, M6zX)`
>
> Per-layer MSO results (**Appendix G, Figures 10-14**) never show alignment and harmful updates as the strongest pair, and layer-localized projections (**Appendix I, Figures 18-19**) fail to isolate safety directions or preserve utility at single layers. Thus, our findings are robust to a layer-wise aggregation strategy, and they argue against the existence of a linearly separable safety subspace.
>
> `Ruling out potential confounds due to refusal and loss in general ability (Reviewers YiuK, M6zX)`
>
> MMLU (general ability) performance stays >60% and refusal rates remain very low across projection settings while harmfulness shifts as expected (**Appendix J, Figures 20-21**). This confirms that our findings are not confounded due to over-refusal or general ability loss.
>
> `Activation-space results over all layers and earlier token windows (Reviewers hV1p, QX9G, M6zX)`
>
> Across all layers and earlier token windows (**Appendix K.1, Figures 22-24, and Appendix K.2, Figure 25**), harmful and benign prompts maintain consistently high subspace overlap. This confirms that no safety-selective subspaces emerge across layers or token windows.
>
> `Activation space results when using a different-style refusal dataset instead of a “useful” one (Reviewers QX9G, M6zX)`
>
> Switching to a refusal-style dataset produces the same overlap pattern (**Appendix K.3, Figure 26**): harmful and benign prompts remain highly overlapped. This confirms that the observed entanglement is robust to the choice of supervision data.
>
> `Comparative discussion with activation steering methods (Reviewer Xrmm)`
>
> **Appendix L** provides a detailed explanation of how our observations compare with the success of activation steering methods.
>
> ---
>
> We believe these updates meaningfully addressed all reviewer concerns and helped clarify and strengthen our paper's claims.

---

### Meta-Review · Area_Chair_sSBV · 2026-01-07

**Summary:**

This paper conducts empirical study to provide insights on the entanglement between safety related behaviour and general purpose learning. The investigation is on linear subspaces of weights and activations.

Few of the major concerns were
* what about non-linear alternatives?
* lacks theoretical investigation
* how the findings depend on the assumption of linear separability hypothesis?
* more experiments using different safety fine-tuning methods

**Reviewer Concerns:**

I believe that most of the major concerns were addressed by the authors except the theoretical justification part. However, I do believe that such empirical studies also add significant value in shaping the thinking process of the community and making process.

**Reviewer Scores:**

The new _minimum_ scores would have been 6, 4, 6, 6, 8 leading to an average score of 6.

---

### Decision · Program_Chairs · 2026-01-26

Accept (Poster)